# Structural convergence endows nuclear transport receptor Kap114p with a transcriptional repressor function toward TATA-binding protein

Chung-Chi Liao[1,2], Yi-Sen Wang[2], Wen-Chieh Pi [3], Chun-Hsiung Wang [4], Yi-Min Wu[4], Wei-Yi Chen [3,5] ✉ & Kuo-Chiang Hsia [1,2,3] ✉

The transcription factor TATA-box binding protein (TBP) modulates gene expression in nuclei. This process requires the involvement of nuclear transport receptors, collectively termed karyopherin-β (Kap-β) in yeast, and various regulatory factors. In previous studies we showed that Kap114p, a Kap-β that mediates nuclear import of yeast TBP (yTBP), modulates yTBP-dependent transcription. However, how Kap114p associates with yTBP to exert its multi-faceted functions has remained elusive. Here, we employ single-particle cryo-electron microscopy to determine the structure of Kap114p in complex with the core domain of yTBP (yTBP$^C$). Remarkably, Kap114p wraps around the yTBP$^C$ N-terminal lobe, revealing a structure resembling transcriptional regulators in complex with TBP, suggesting convergent evolution of the two protein groups for a common function. We further demonstrate that Kap114p sequesters yTBP away from promoters, preventing a collapse of yTBP dynamics required for yeast responses to environmental stress. Hence, we demonstrate that nuclear transport receptors represent critical elements of the transcriptional regulatory network.

TATA box-binding protein (TBP) is a key protein that facilitates transcription of ribosomal RNAs (rRNAs), messenger RNAs (mRNAs), and transfer RNAs (tRNAs) by the three eukaryotic RNA polymerases: Pol I, Pol II and Pol III, respectively[1–3]. Notably, TBP is required for transcription initiation, as it recognizes the conserved AT-rich sequences at core promoters and mediates the assembly of the pre-initiation complex (PIC)[1,2,4]. Regardless of the species in which it was identified, full-length TBP typically contains two regions[5–7]; its N-terminal unstructured segment varies in length and is dispensable for transcription[8,9], but its C-terminal domain of ~180 amino acids (a.a.) is highly conserved across different species and is essential for transcription activation[6,7,10].

The C-terminal domain consists of two pseudo-symmetrical repeats, each of which contains five β-strands and two α-helices. Moreover, crystal structures of the conserved core domain have revealed a saddle-like structure with convex and concave surfaces[6,7,10]. The concave hydrophobic surface of TBP interacts with the minor groove of the DNA, whereas the highly charged convex surface mediates binding with diverse transcription regulators[6,7,10].

In eukaryotes, the nuclear envelope (NE) physically separates nuclear transcription and cytoplasmic translation. Given that TBP is synthesized in the cytoplasm, it must be imported into the nucleus to exert its functions in transcription. Nuclear pore complexes (NPCs)

[1]Molecular and Cell Biology, Taiwan International Graduate Program, Academia Sinica and National Defense Medical Center, Taipei 11490, Taiwan. [2]Institute of Molecular Biology, Academia Sinica, Taipei 11529, Taiwan. [3]Institute of Biochemistry and Molecular Biology, College of Life Sciences, National Yang Ming Chiao Tung University, Taipei 11221, Taiwan. [4]Institute of Biological Chemistry, Academia Sinica, Taipei 11529, Taiwan. [5]Cancer and Immunology Research Center, National Yang Ming Chiao Tung University, Taipei 11221, Taiwan. ✉e-mail: chenwy@nycu.edu.tw; khsia@gate.sinica.edu.tw

represent the only channels through the NE. However, TBP undergoes self-assembly to form a homo-dimer, increasing its molecular weight beyond the diffusion limit (~40 kDa) of NPCs and thereby greatly limiting its passive diffusion-dependent nuclear localization[5,11]. A group of nuclear transport receptors, collectively termed karyopherin-β (Kap-β) in yeast, facilitates active nucleo-cytoplasmic transport[12]. Kap-βs recognize either nuclear localization signals (NLSs) or nuclear export signals (NESs) located in the polypeptide chains of the cargoes, and transfer them from one side of the NE to the other. Notably, Kap-β-cargo interaction and transport directionality are governed by Ran GTPases[12,13]. Multiple Kap-βs mediate active nuclear import of yeast TBP (yTBP) (e.g., Kap95p, Kap114p, Kap121p and Kap123p)[14,15]. It is not clear why yTBP is recognized by multiple Kap-βs, with many Kap-βs showing redundancy in their nuclear import of yTBP. One possibility is that different Kap-β-yTBP complexes facilitate distinct cellular functions. Hence, whether or not the Kap-β-yTBP complex employs different structural features to carry out functions beyond nuclear transport needs to be established.

Notably, our previous study revealed that Kap114p exhibits a sub-nanomolar affinity for the core domain of yTBP (a.a. 61-240; hereafter referred to as yTBP$^C$), four orders of magnitude greater than its counterparts, and suppresses binding of yTBP$^C$ with DNA[16]. We hypothesized that the exceedingly strong binding affinity of Kap114p for yTBP, more than what is required for nuclear import, may facilitate additional functions. Moreover, given that Kap114p can repress gene expression, particularly under high salt stress, we proposed a Kap114p-mediated suppression pathway that regulates stress-responsive gene expression[16]. However, whether or not Kap114p modulates the yTBP-promoter interaction in vivo, and thus participates in the yTBP regulatory network, is unclear. Structural information at near-atomic resolution for Kap114p in complex with yTBP is required to elucidate the molecular details underlying the strong binding affinity between Kap114p and yTBP$^C$ that accounts for Kap114p's suppressive function.

A variety of transcriptional regulatory proteins interact with TBP to modulate its binding to promoters[1,6,17–21]. Thus, competition among TBP's interacting partners alters the dynamics of TBP-containing complex assembly and thereby modulates transcription initiation to affect gene expression[22,23]. Much effort has been exerted to understand the interplay among transcription factors (e.g., transcriptional regulators such as TATA-box binding protein associated factor 1 (TAF1)[19,24] and regulatory proteins (e.g., modifier of transcription 1 (Mot1)[17,21]), and how they regulate TBP-dependent transcription. However, nuclear transport receptors have classically not been considered transcriptional regulators and so they are typically not included in the transcriptional regulatory network. Thus, a complete assessment of the functions of a transcriptional regulatory network cannot be achieved when an important group of players is excluded.

Here, we used single-particle cryo-electron microscopy (cryo-EM) to determine the structure of the Kap114p•yTBP$^C$ complex at a resolution of ~4 Å. The cryo-EM structure reveals that Kap114p wraps around the N-terminal lobe of yTBP$^C$, mainly via three binding patches, and thus structurally resembles how TAF1 and Mot1 form a complex with TBP. A short α-helix in the Kap114p HEAT19 loop binds to the hydrophobic concave surface of yTBP$^C$ in a similar way to the TAF N-terminal domain (TAND) of TAF1 and a latch region of Mot1[21,24]. Moreover, multiple signature residues located in the N-terminal lobe of TBP and that interact with several TBP-associated regulatory proteins (e.g., Mot1[23]) also contribute to Kap114p binding. The structural information we present demonstrates that although the overall three-dimensional structures of Kap114p and TBP-associated regulatory proteins differ, they interact with the same regions of TBP, implying convergent evolution of these two groups of proteins. Yeast genetics and ChIP-seq analyses further corroborate

that Kap114p serves as a yTBP suppressor in the transcriptional regulatory network. Moreover, we show that Kap114p-dependent sequestration prevents a collapse in the yTBP dynamics required for yeast to respond appropriately to environmental stress (e.g., salt stress).

## Results

### Single-particle cryo-EM of the Kap114p•yTBP$^C$ complex

Previous biochemical analyses revealed a stronger binding affinity for Kap114p to yTBP$^C$ compared to that of other Kap-βs[16]. Based on that finding and the results of transcriptome analysis, we proposed a regulatory function for Kap114p in yTBP-mediated gene transcription, in addition to its nuclear transport activity[16]. To gain structural insights into how Kap114p binds to yTBP$^C$ with an affinity in the sub-nanomolar range to exert its suppressive function, we used cryo-EM to determine the structure of the Kap114p•yTBP$^C$ complex.

To prepare homogeneous samples, we purified individual complex components from *E. coli* and then assembled the Kap114p•yTBP$^C$ complex by mixing Kap114p and yTBP$^C$ at an equal molar ratio, followed by cross-linking with different concentrations of glutaraldehyde (Supplementary Fig. 1a). Cross-linked protein samples in the presence of 0.01% glutaraldehyde were further purified by size exclusion chromatography (Supplementary Fig. 1b). The peak fractions were subsequently vitrified on cryo-EM grids (Supplementary Fig. 1b), and micrographs were acquired using a Titan Krios microscope (300 keV) (Supplementary Fig. 1c). Two-dimensional (2D) class averaging revealed secondary structural elements and recognizable structural features of the Kap114p•yTBP$^C$ complex (Supplementary Fig. 1d). Next, multiple rounds of three-dimensional (3D) reconstruction were performed using the best-resolved classes across each classification (Supplementary Fig. 2a). Ultimately, a cryo-EM structure of the Kap114•yTBP$^C$ complex, representing a -130 kDa heterodimeric protein complex, was determined to an average resolution of 4.03 Å using 212,899 particles (Supplementary Fig. 2a, b and Table 1). The highest local resolution we achieved was 3.5 Å for the core region (Supplementary Fig. 2c, d), allowing us to validate residue identity in the protein-protein interactions.

In the cryo-EM maps, we identified a saddle-shaped density exhibiting two-fold symmetry that is enwrapped by a curved density displaying rod-like features characteristic of tandem repeats (Fig. 1a). Based on structures determined by X-ray crystallography, yTBP$^C$ is a symmetrical structure comprising two repeated domains (N- and C-lobes), and Kap114p displays a solenoid-like structure containing 20 tandem HEAT repeats[7,16]. Hence, we could unambiguously assign the orientations of the yTBP$^C$ and Kap114p crystal structures into the saddle-shaped and armadillo-like densities, respectively, based on the structural features resolved in the cryo-EM maps (Supplementary Fig. 3a). Moreover, we identified an additional density next to the concave surface of yTBP$^C$ (Fig. 1a, b). Protein secondary structure prediction combined with previously biochemical studies enabled us to assign an α-helix of HEAT 19 loop to that density (Figs. 1b−e and 2i)[16].

In the Kap114•yTBP$^C$ complex, Kap114p is arranged into a right-handed superhelix and forms a 1:1 stoichiometric complex with yTBP$^C$ (Fig. 1c, d). Notably, Kap114p mainly wraps around the pseudosymmetric N-terminal lobe of yTBP$^C$, primarily via three binding patches (the B α-helix and loop of HEAT8, the B α-helix of HEAT13, and the α-helix of the HEAT19 loop; highlighted by light blue, light green and light orange in Fig. 1c−e). Significantly, the three binding interfaces of Kap114p that contribute to the Kap114p•yTBP$^C$ interaction partially overlap with its RanGTPase-binding regions (Supplementary Fig. 3b)[13]. Thus, binding of Ran to Kap114p only partially dissociates yTBP from Kap114p[15,16]. Additionally, the enwrapment mediated by multiple binding sites of Kap114p may account for blockage of yTBP dimerization and yTBP-DNA interaction[5,7,11,16].

**Table 1 | Cryo-EM data collection, refinement, and validation statistics**

|  | Kap114p•yTBP$^C$ (EMD-34490) (8H5B) |
|---|---|
| *Data collection* |  |
| EM equipment | Titan Krios |
| Voltage (kV) | 300 |
| Cs (mm) | 2.7 |
| Magnification (nominal) | 105,000 |
| Detector | K3 |
| Pixel size (Å) | 0.83 |
| Electron exposure (e⁻/Å²) | ~40 |
| Exposure time (s) | 2.5 |
| Frames (no.) | 40 |
| Defocus range (µm) | −1.5 to −2.5 |
| *Reconstruction* |  |
| Software | Relion & cryoSPARC |
| Micrographs stacks (no.) | 6,807 |
| Final particle images (no.) | 212,899 |
| Symmetry imposed | C1 |
| Map final resolution (Å)[a] | 4.03 |
| Map sharpening B-factor (Å2) | -294.0 |
| *Atomic modeling* |  |
| Software | Coot & Phenix |
| Number of protein residues | 1075 |
| Number of ligands | 0 |
| Number of atoms | 8570 |
| Mask CC[b] | 0.7365 |
| Volume CC[b] | 0.7212 |
| RMSD bond lengths (Å) | 0.005 |
| RMSD bond angles (°) | 0.680 |
| Clash score[b] | 22.19 |
| Ramachandran favored (%)[b] | 95.46 |
| Ramachandran allowed (%)[b] | 4.54 |
| Ramachandran outliers (%)[b] | 0.00 |
| Rotamer outliers (%)b | 0.00 |
| C$_\beta$ deviations[b] | 0.00 |
| MolProbity score[b] | 2.16 |
| EMRinger score | 0.53 |

[a]According to FSC = 0.143.
[b]According to the criterion of Chen et al.[52].

## An α-helix of the HEAT19 loop in Kap114p binds to the concave surface of yTBP in a similar fashion to TAF1 and Mot1

The human homolog of Kap114p, Importin-9, in complex with histones H2A-H2B heterodimer has recently been determined using X-ray crystallography[25]. Although multiple binding sites (e.g., N-terminal loops, HEAT13 B and the HEAT18-19 loop) contribute to the interaction of H2A-H2B and Importin-9, H2A-H2B and yTBP$^C$ bind at different areas of Importin-9 and Kap114p, respectively (Supplementary Fig. 3c). The root mean square deviation (rmsd) of 6.4 Å (Cα; as assessed in Pymol) between the Kap114p and Importin-9 structures also underscores the significant differences between them (Supplementary Fig. 3d). This divergence is particularly striking in the C-termini of the solenoid structures, implying considerable structural rearrangement to selectively interact with TBP and H2A-H2B (Supplementary Fig. 3d). Moreover, in Importin-9, the HEAT18-19 loop, but not the HEAT19 loop, mediates the H2A-H2B dimeric interaction[25] (Supplementary Fig. 3c). However, isothermal titration calorimetry (ITC) using a Kap114p deletion mutant lacking the HEAT19 loop (a.a. 932-941) revealed a binding affinity to yTBP$^C$ ~100-fold lower than determined for the wild type (~158 nM versus ~1 nM) (Fig. 2a, b; Kd values of wild type and mutants are listed in Supplementary Table 1). The loop located between HEAT18 and HEAT19 at the convex surface of Kap114p exerted no influence on yTBP$^C$ interaction (Fig. 2c). These results indicate that the HEAT19 loop of Kap114p, but not its HEAT18-19 loop, significantly contributes to yTBP$^C$ binding. Deletion of residue Y939 (inside the α-helix of the HEAT19 loop) resulted in greater perturbation of the Kap114p•yTBP$^C$ complex binding affinity compared to deletion of residue Y933 (outside of the α-helix of the HEAT19 loop) (Fig. 2d, e, i), corroborating that the α-helix in HEAT19 is critical for yTBP binding and further validating our protein sequence assignment. Thus, the conformation by which Kap114p wraps around the N-terminal surface of yTBP$^C$ exhibits minimal similarity to previously reported Importin-9-H2A-H2B complex structures[25].

Notably, we did not observe any large conformational changes for either Kap114p or yTBP$^C$ between the cryo-EM and previously reported crystal structures. For instance, the yTBP$^C$-bound and -unbound Kap114p structures display no apparent structural differences (Cα rmsd = 1.7 Å, as assessed in Pymol), apart from the HEAT19 loop that is only assigned in yTBP$^C$-bound Kap114p and not in the crystal structure of Kap114p alone (Supplementary Fig. 3e). Moreover, the TBP structures in the Kap114p-bound, Mot1-bound, DNA-bound and homo-dimeric forms are highly similar (Cα rmsd = 0.9 to 1.1 Å, as assessed in Pymol) (Supplementary Fig. 3f). Hence, Kap114p binding does not induce substantial conformational changes in yTBP$^C$, consistent with our understanding of how other TBP-associated factors promote TBP remodeling as a consequence of binding without changing the TBP structure[23]. Additionally, Kap114p has been shown to interact with the Nap1p•H2A•H2B complex[26,27], but that interaction only displayed an exothermic curve and a dissociation constant (Kd) value of ~6.8 µM (Supplementary Fig. 4a). Thus, the Kap114p•Nap1p•H2A•H2B complex displays a greater Kd value, in the micro-molar range, compared to the Kap114p•yTBP$^C$ interaction (1 nM) (Fig. 2a). However, this binding strength is comparable to that of yTBP binding to Kap95p or Kap121p[16]. These data indicate that different cargo binding modes with variable binding affinities co-exist in a nuclear transport receptor and thereby facilitate distinct cellular functions.

Furthermore, TAF1 and Mot1 employ structural motifs that use α-helices to occupy the DNA-binding concave surface of TBP, modulating binding of TBP to DNA (Fig. 2f and Supplementary Fig. 5a–d)[21,24]. Essentially, the Kap114p-HEAT19 loop, TAF1-TAND and Mot1-latch mainly bind to the N-lobe of TBP (Fig. 2f). The structural configuration that facilitates TBP binding of these α-helical motifs are distinctive: (1) the buried surface areas between TAF1-TAND (886 Å²) or Mot1-latch (1180 Å²) and TBP are much larger compared to that of the Kap114p-HEAT19 loop (375 Å²) (calculated using the software Protein Interfaces, Surfaces and Assemblies (PISA))[28]; and (2) the TAF1-TAND and Mot1-latch exhibit a similar reverse sequence orientation to interact with TBP[21,24]. The α-helix of the Kap114p-HEAT19 loop lies perpendicular to the TAF1-TAND and Mot1-latch motifs (Fig. 2f). Notably, multiple conserved residues (F99 and Q158) in the concave surface of yTBP facilitate interactions with α-helices in the HEAT19 loop of Kap114p, TAF1-TAND and Mot1-latch (Fig. 2f-i)[21,24]. Residue Y939 of Kap114p interacts with yTBP$^C$ Q158, just as F23 of TAF1-TAND1 binds to Q158 of yTBP and F123 of Mot1 binds to Q116 of *Ec*TBP (Fig. 2g–i)[21,24]. Residues Y933 of Kap114p and F99 of yTBP$^C$ facilitate hydrophobic interactions, comparable to those of Mot1 F129 to *Ec*TBP F57/F74 and TAF1-TAND1 Y19 to yTBP F99/F116 (Fig. 2g–i)[21,24]. These results demonstrate that although the Kap114p-HEAT19 loop, TAF1-TAND1 and Mot1-latch use distinct structural features to interact with TBP, they share interactions with multiple TBP residues.

## The convex surface of the yTBP N-lobe mediates interactions of Kap114p with TBP regulatory proteins

The N-terminal lobe of TBP serves as a platform to enable interaction with multiple TBP-associated factors, including general transcription

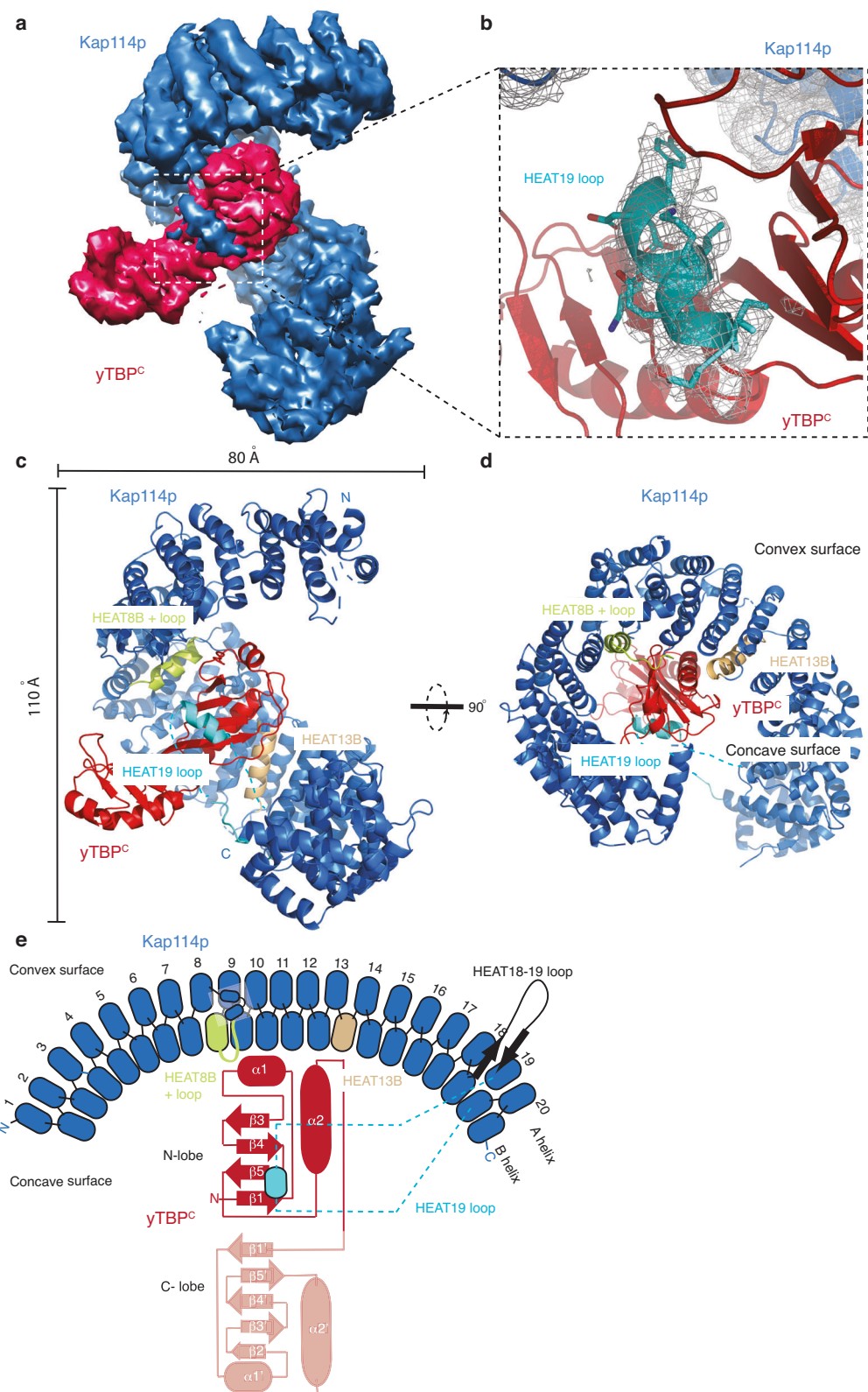

**Fig. 1 | Structural characterization of the Kap114p•yTBP^C complex. a** A cryo-EM map of the Kap114p•yTBP^C complex. The map is segmented into Kap114p (blue) and yTBP^C (red). **b** The map of Kap114p with a superimposed atomic model of the Kap114p•yTBP^C complex, showing Kap114p in blue and yTBP^C in red. The HEAT19 loop of Kap114p is highlighted in light blue. The side chains of HEAT19 loop residues 932 to 941 are shown as sticks. **c** Cartoon representation of the Kap114p•yTBP^C structure. Kap114p and yTBP^C are shown in blue and red, respectively. The HEAT8 loop, HEAT13B and HEAT19 loop that contribute to the yTBP^C interaction are highlighted in light green, light orange and light blue, respectively. Overall dimensions of the structure are indicated. **d** A 90° rotated view of c. The HEAT8 loop, HEAT13B and HEAT19 loop are indicated. **e** Schematic representation of the Kap114p domain configuration using the same color code as shown in (**c**). The N- and C-lobes of yTBP^C are shown in dark and light red, respectively. The HEAT18/19 loop that mediates histones H2A and H2B binding is shown in black[25].

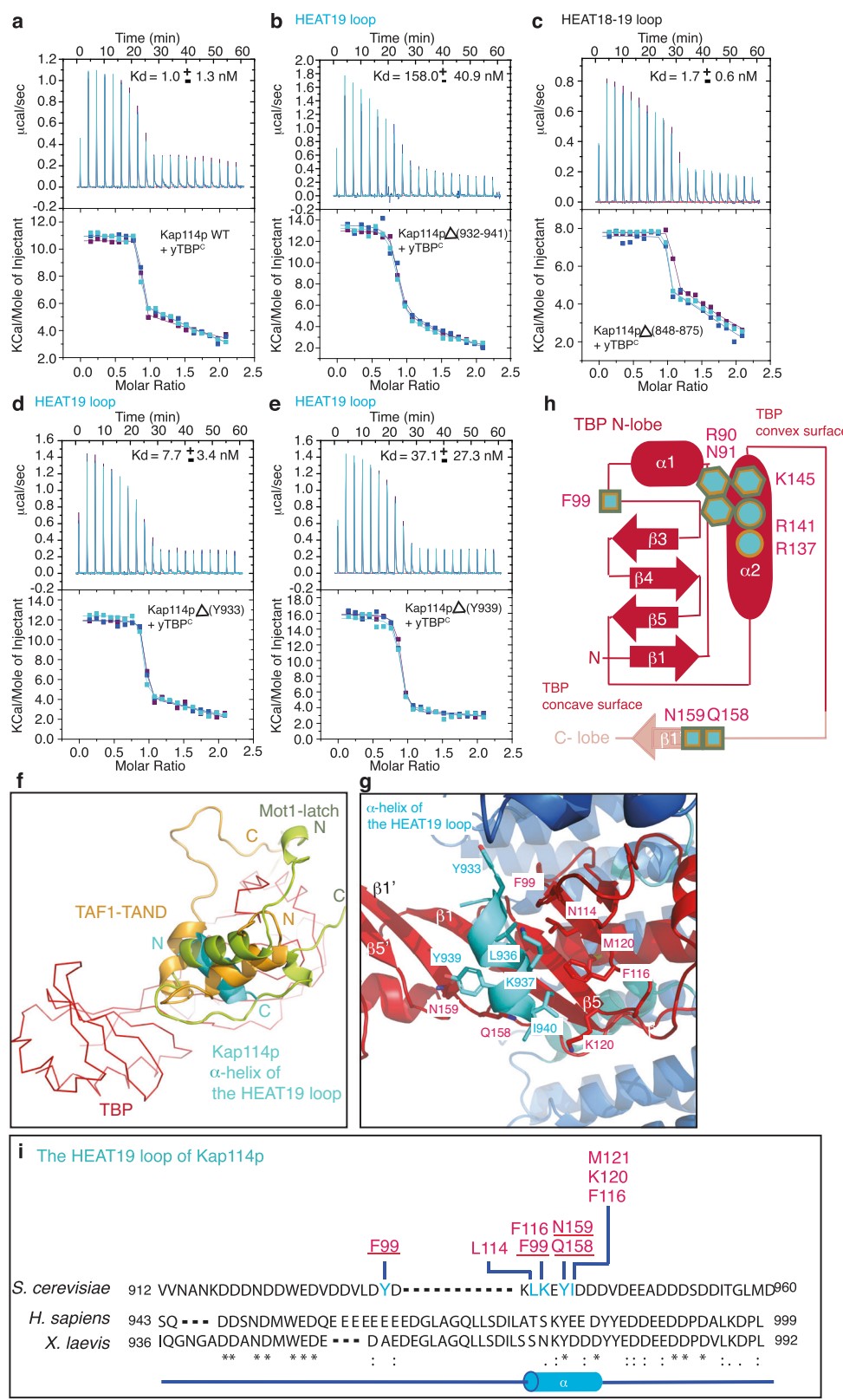

factors and TBP regulatory factors[23]. Therefore, we superimposed our cryo-EM model of the Kap114p•yTBP[C] complex on the previously resolved structures of the Mot1-NTD•TBP and TAF1-TAND•TBP complexes, which revealed a remarkably comparable interacting mode among these protein pairs (Fig. 3). Among the residues in the convex surface involved in the interaction between TBP and TBP-associated factors (Fig. 3c, d; highlighted in pink), five signature residues that

were identified previously and are spatially clustered yet far from Pol II transcription machinery-binding regions also interact with Kap114p (Fig. 2h)[23]. Among these five residues, R90 and N91, located near the α-helix 1 of yTBP[C], are bound by the HEAT8 loop of Kap114p (Figs. 2h, 4a and Supplementary Fig. 5e) and exhibit close contacts with the HEAT8 loop via residues S369 and Y370 (Fig. 4a). The other three signature residues—R137, R141 and K145—are located at α-helix 2 of yTBP[C] and

**Fig. 2 | A short α-helix in the HEAT19 loop of Kap114p mediates its binding to the concave surface of yTBPᶜ. a–e** ITC titration curves (upper) and binding isotherms (lower) of wild type (**a**), Kap114pΔ(932-941) (**b**) Kap114pΔ(848-875) (**c**), Kap114pΔY933 (**d**) and Kap114pΔY939 (**e**) with yTBPᶜ. Kd values are indicated. Data are represented as mean ± SD (*n* = 3 independent experiments). **f** Superimposition of the α-helix of the HEAT19 loop (light blue) with TAF1-TAND (yellow) and the Mot1 latch motif (light green). yTBPᶜ is shown in red. The N- and C-termini of each peptide are indicated. **g** A close up view showing the interaction between the α-helix of the HEAT19 loop (light blue) and the concave surface of yTBPᶜ (red). Key residues involved in the binding are indicated. yTBPᶜ and the α-helix are shown as a stick with line side-chain and cartoon representations, respectively. **h** Schematic representation of the N-lobe of yTBPᶜ. Conserved residues that mediate Kap114p,

TAF1 and Mot1 interactions in the TBP concave and convex surfaces are indicated by squares, hexagons and circles, respectively. The same color code as shown in B has been used. **i** Alignment (by Clustal Omega[53]) of protein sequences of the HEAT19 loop from indicated species (numbers represent amino acid positions). * indicates fully conserved residues;: represents residues showing strongly similar properties;. indicates residues displaying weakly similar properties. An α-helix was assigned to a segment of the amino acid sequence based on protein secondary structure prediction (red, as assessed in PSIPRED[54]) and is depicted as a light blue cylinder. Residues involved in the inter-molecular interaction network of the Kap114p HEAT19 loop (light blue) in complex with yTBPᶜ (red) are indicated. Blue lines indicate interaction residues.

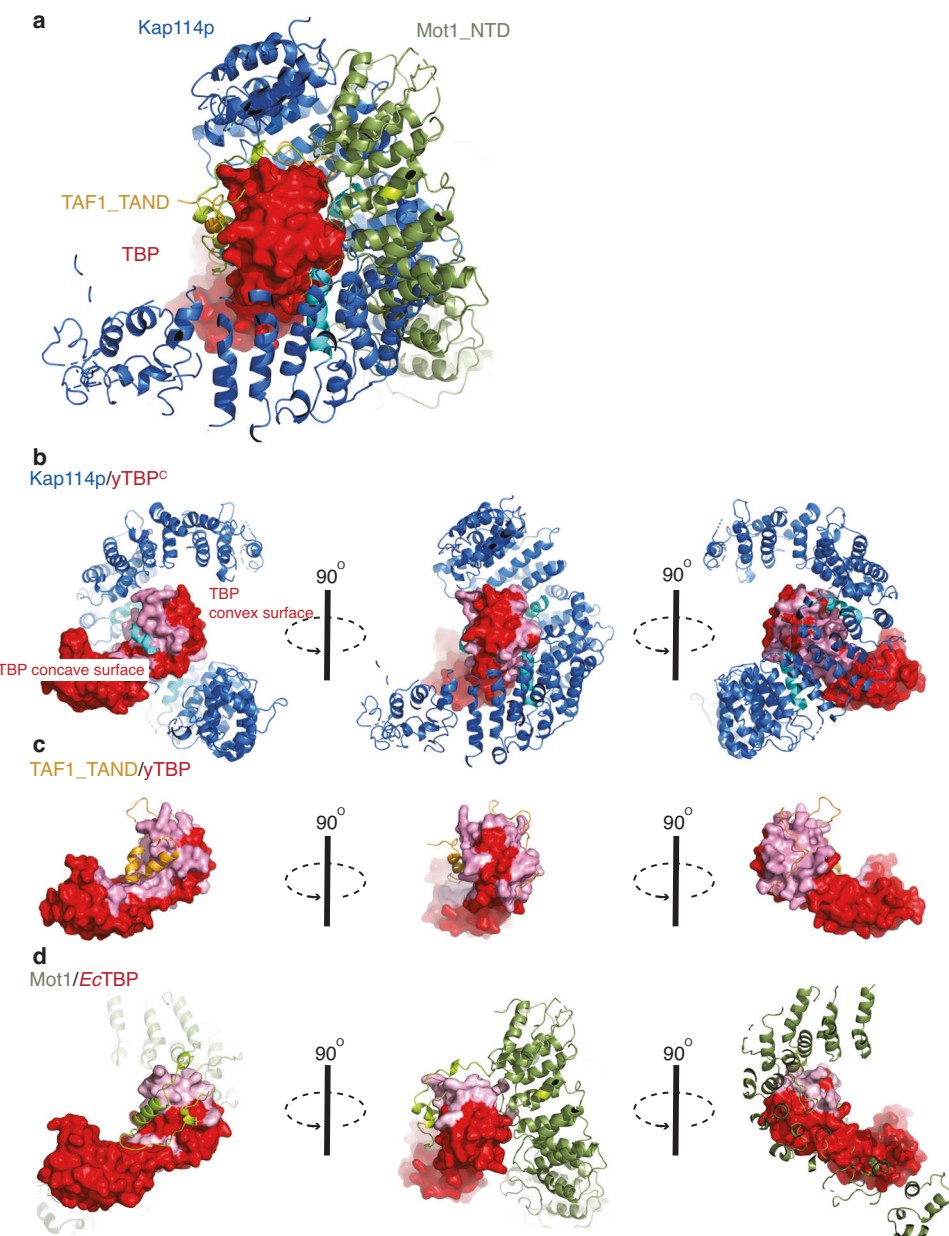

**Fig. 3 | Kap114p, TAF-TAND and Mot1-NTD share common structural features in wrapping around the N-lobe of yTBPᶜ. a** Superimposition of the Kap114p (blue), TAF1-TAND (yellow) and Mot1-NTD (green) cartoon representations. TBP is shown in red as a surface representation. **b–d** Protein complex structures of

Kap114p•yTBPᶜ (**b**), TAF1-TAND•yTBP (**c**), and Mot1-NTD•*Ec*TBP (**d**), with Kap114p, TAF1-TAND and Mot1-NTD being shown in cartoon representations and all TBPs as surface representations. Three rotated views of each structure are shown. Interacting surfaces of each structure are shown in pink.

interact with the HEAT8 loop and HEAT13B (Figs. 2h, 4b and Supplementary Fig. 5g). Moreover, they form multiple hydrogen bonds and salt bridges with Kap114p residues, such as N591 and Q597 (Figs. 2h, 4b and Supplementary Fig. 5f).

To validate if these five signature residues in yTBP$^C$ contribute to its interaction with Kap114p at two interfaces (B-helix and loop of HEAT8 and HEAT13B), we mutated them (Fig. 4a, b) and used ITC to measure the Kd values for the mutants. Wild type Kap114p and yTBP$^C$ displayed a high binding affinity (Kd value of ~1 nM; Fig. 2a)[16]. Notably, under the same ITC conditions, single point mutations of yTBP$^C$ (N91A, R141A and K145A) and Kap114p (Y370A and Q597A) at their binding interfaces resulted in weakened binding relative to wild type (Fig. 4c, d, f, g, h). Moreover, double point mutations of residues in Kap114p•yTBP$^C$ interacting surfaces (Kap114p(Y370A)+yTBP$^C$(N91A) and Kap114p(Q597A)+ yTBP$^C$(R141A)) greatly diminished binding between the Kap114p and yTBP$^C$ components (Fig. 4e, i). In particular, the double mutant combining Y370A on Kap114p and N91A on yTBP$^C$ resulted in a Kd value of ~1.5 μM for the complex, i.e., more than a thousand-fold weaker than wild type (Figs. 2a, 4e). Notably, these results were further validated at lower protein concentrations (25 μM for Kap114p and 250 μM for yTBP$^C$), as well as for buffer controls (Supplementary Table 1 and Supplementary Fig. 4b–g). Taken together, these biochemical results corroborate that the convex surface of the yTBP N-lobe mediates the Kap114p and yTBP$^C$ interaction.

In addition to Kap114p, various other Kap-βs (e.g., Kap95p) have been proposed to facilitate nuclear import of yTBP[14,15], and ITC analysis revealed that the binding constants between yTBP$^C$ and these Kap-βs are in the sub-micromolar range[16]. For example, R141A mutation of yTBP$^C$ elicited a >200-fold reduction in the Kap114p-yTBP$^C$ binding affinity (Figs. 2a, 4h), whereas this same mutation only resulted in a minor decrease of its binding affinity for Kap95p (~9.7 μM: comparable to that of the wild type at ~8.1 μM) (Fig. 4j, k). Thus, yTBP$^C$ likely binds Kap114p and Kap95 differently, resulting in distinct interaction strengths and enabling its diverse cellular functions. Taken together, the structural features of Kap114p imply convergent evolution of two distinct protein groups, i.e., nuclear transport receptors (e.g., Kap114p) and transcriptional regulators (e.g., Mot1). Therefore, we propose that Kap114p may function as a TBP regulatory protein, playing an important role in the transcriptional regulatory network.

## Kap114p modulates the genomic DNA binding of yTBP

Our cryo-EM study has revealed that Kap114p, TAF1 and Mot1 share conserved binding patches on TBP, prompting the hypothesis that Kap114p can function as a TBP regulator by competing for TBP interaction with TAF1 and Mot1. To explore that possibility, we examined the interplay between Kap114p and TAF1 in yeast grown under high salt stress. We have previously demonstrated that, through its negative regulation of yTBP activity, Kap114p is required for yeast to grow under chronic salt stress[16]. Similarly, knockout of KAP114 in yeast resulted in a growth defect relative to wild-type strains solely under high salt conditions, as revealed by yeast spot-based assays and yeast growth curves (Fig. 5a–c and Supplementary Fig. 6a)[16]. Notably, the WT and KAP114 knockout mutants showed differences both in the growth rate during the exponential stage and the final total biomass under high salt conditions (Fig. 5b, c). One cannot exclude the possibility that KAP114 knockout can lead to growth defects in both cell growth rate and viability.

Given that TAF1 can facilitate activation of TBP-dependent transcription[24,29], deletion of the TBP-interacting domain of TAF1 (Δtaf1-tand) in yeast elicited a more severe growth defect than observed for wild-type or KAP114 single-knockout strains under the condition of high salt treatment (Fig. 5a–c and Supplementary Fig. 6a). Remarkably, simultaneous depletion of both KAP114 and TAF1-TAND partially rescued the defective growth phenotypes caused by sole TAF1-TAND disruption under high salt conditions (Fig. 5a–c and

Supplementary Fig. 6a), indicating that Kap114p is part of the transcription network and likely functions as an antagonist to the activity of TAF1, and that it is particularly critical under high salt stress (Fig. 5d).

To further investigate if Kap114p plays a role in regulating genomic DNA binding by yTBP, we profiled genome-wide yTBP occupancies in KAP114 knockout strains rescued with wild-type or mutant KAP114 (KAP114(Δ899-956)) under regular and high-salt conditions. Given that available antibodies against yTBP are not efficient in chromatin immune-precipitation sequencing (ChIP-seq) assays, we used yeast strains expressing GFP-tagged yTBP[30]. To examine the transcriptional function of GFP-tagged yTBP, we determined the gene expression mediated by RNA Pol I (RDN58), RNA Pol III (SNR6), and RNA Pol II (TFIID-dependent RPS5, RPS8A, RPS3; SAGA-dependent PYK1 and PGK1)[31]. Although the GFP-yTBP-bearing strain grew slower than its parental strain (BY4741, Supplementary Fig. 6b, c), we detected no significant differences in expression from RT-qPCR data for the seven TBP-driven genes in the strain expressing GFP-tagged yTBP and its parental strain (BY4741) under regular and high-salt conditions (Supplementary Fig. 6d, e). These results highlight that GFP tagging of yTBP does not perturb its transcriptional activity. Although the strain expressing GFP-tagged yTBP grew slower than its parental strain, this outcome could be attributable to indirect effects on pathways other than those responsible for transcriptional regulation. Notably, our ChIP-seq experiments were carried out under the same conditions, so the results are comparable. Moreover, a Western blot analysis revealed generally comparable protein expression levels of GFP-yTBP among the three yeast strains we examined under regular and high-salt conditions (Supplementary Fig. 6f–h). RT-qPCR analysis further corroborated similar yTBP and KAP114 transcript levels among the different strains (Supplementary Fig. 6i, j).

ChIP-seq profiling with an anti-GFP antibody revealed that GFP-tagged yTBP bound to the core promoters of 2,284 protein-coding genes (mRNA) and 272 tRNA genes in the KAP114 knockout strain (tag density > 100 rpkm and TBP ChIP-seq signal > 2-fold enrichment over input DNA). Expression of wild-type KAP114, but not a HEAT19 loop deletion mutant, resulted in a statistically significant reduction of GFP-yTBP binding at the promoters of these genes under regular conditions (Fig. 5e, f). Our previous study showed that the HEAT19 loop deletion mutant can rescue the mislocalization of yTBP elicited by KAP114 knockout[16]. The results shown here indicate that global suppression of yTBP-promoter interaction by Kap114p involves its HEAT19 loop, which is not attributable to perturbation of the nuclear yTBP concentration. We confirmed the ChIP-seq data by conducting an independent biological repeat from a separate transformation colony (1,480 mRNA and 272 tRNA genes; tag density > 100 rpkm and 3-fold enrichment of TBP ChIP-seq signal) (Supplementary Fig. 7a, b). Furthermore, we present scatterplots illustrating the correlation of pairwise samples under regular and high-salt conditions in these two ChIP-seq datasets (Supplementary Fig. 7c–n). Pearson correlation coefficients of each pair indicate that the ChIP-seq results display high (mRNA) or moderate (tRNA) reproducibility. We selected 18 genes to further assess by RT-qPCR the yeast strains expressing yTBP with or without the GFP tag (Supplementary Fig. 8a–f). The strong correlation in gene expression for these 18 genes in these strains also supports that the GFP tag does not affect the transcriptional function of yTBP.

Although yeast hosts ~150 ribosomal RNA (rRNA) genes in multiple highly repetitive gene clusters[32], which preclude them from being reliably analyzed by ChIP-seq, we still observed a reduction of GFP-yTBP binding at the rRNA gene RDN5-1 in strains hosting exogenous wild-type KAP114, but not the respective HEAT19 loop deletion mutant (Supplementary Fig. 8g). Thus, our findings indicate that Kap114p may trap yTBP and restrain it from binding to gene promoters under regular conditions.

Furthermore, we also observed that yTBP promoter binding was globally abrogated at mRNA and tRNA genes in KAP114

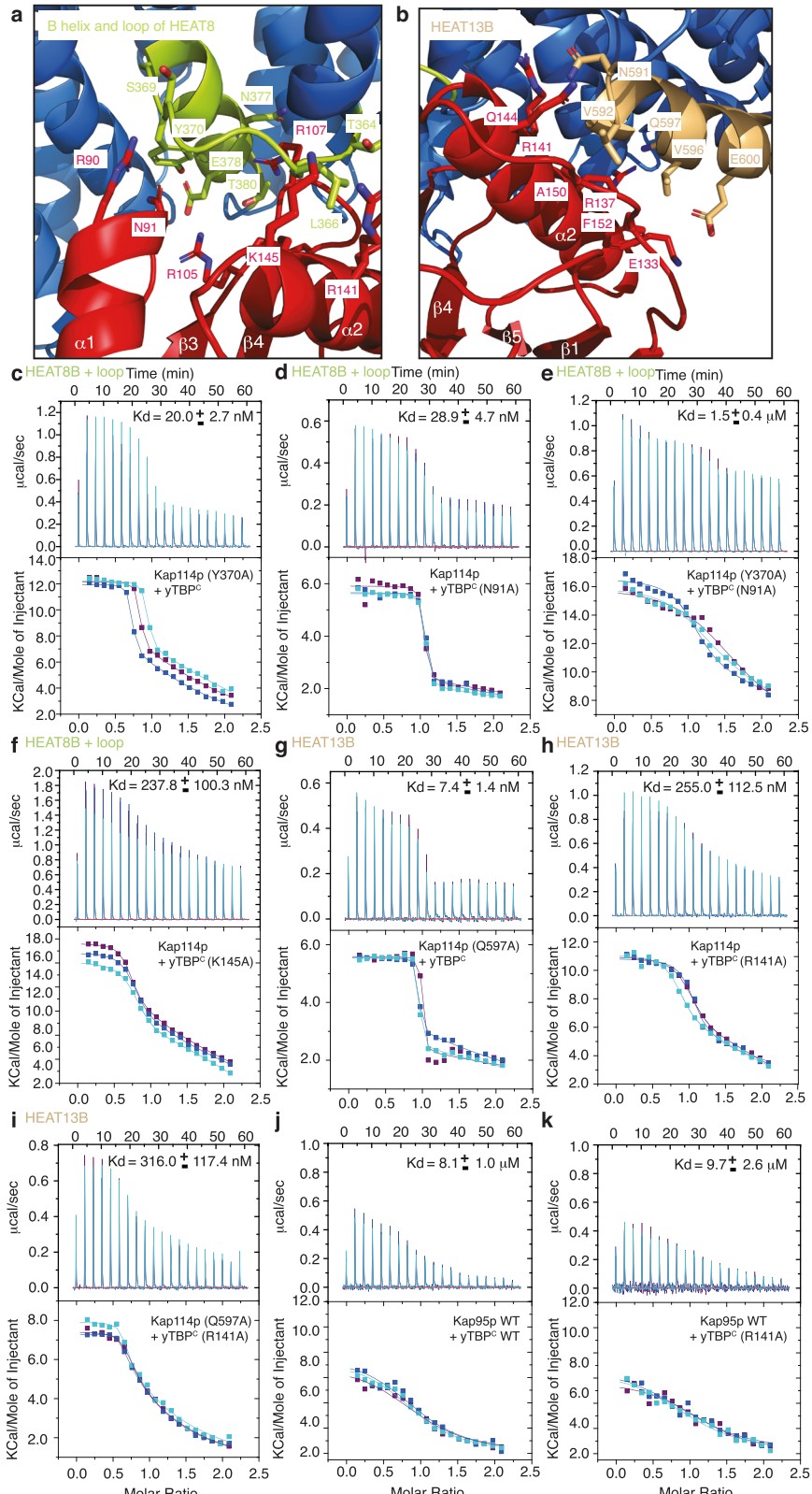

**Fig. 4 | Kap114p interacts with the convex surface of yTBP$^C$. a, b** The two panels highlight the interactions of the B-helix and loop of HEAT8 (**a**) and HEAT13B (**b**) of Kap114p with yTBP$^C$. Residues mediating Kap114p and yTBP$^C$ are indicated. yTBP$^C$ (red) and Kap114p (blue) are shown as a stick with line side-chain and cartoon representations, respectively. The B-helix and loop of HEAT8 and HEAT13B are colored light green and light orange, respectively. **c–i** The panels show ITC titration curves (upper) and binding isotherms (lower) of wild type Kap114p (**d, f, g**), Kap114p

(Y370A) (**c**) or Kap114p (Q597A) (**g**) with wild type yTBP$^C$ (**c**), yTBP$^C$ (N91A) (**d, e**), yTBP$^C$ (K145A) (**f**), or yTBP$^C$ (R141A) (**h, i**). Kd values are indicated. Data are represented as mean ± SD ($n = 3$ independent experiments). **j, k** ITC titration curves (upper) and binding isotherms (lower) of Kap95p with wild type yTBP$^C$ (**j**) or yTBP$^C$ (R141A) (**k**). Kd values are indicated. Data are represented as mean ± SD ($n = 3$ independent experiments).

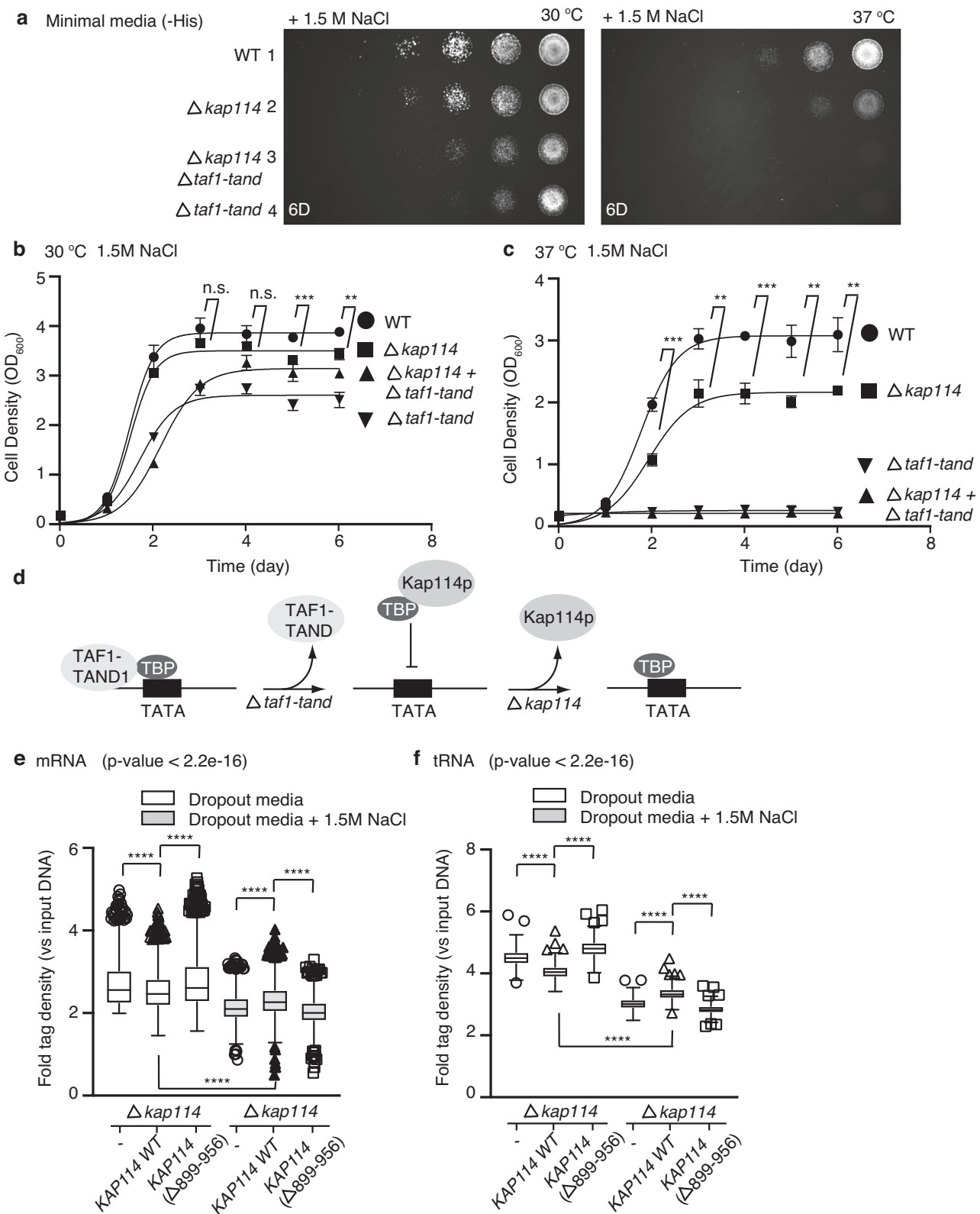

**Nature Communications** | (2023)14:5518

knockout strains under saline stress (Fig. 5e, f). This finding suggests that saline stress may activate robust negative pathways in addition to a Kap114p-dependent pathway to suppress the DNA binding activity of TBP. Remarkably, we found that wild type *Kap114*, but not the HEAT19-deletion mutant, showed a statistically significant enhancement of yTBP binding at the promoters of mRNA and tRNA genes under saline stress (Fig. 5e, f and Supplementary

Fig. 7a, b). To determine if Kap114p is selectively recruited to these genes, we performed ChIP-qPCR assays on yeast strains expressing GFP-tagged Kap114p in the presence of saline stress. Our results show that there is no enrichment of Kap114p binding at selected tRNA promoters (Supplementary Fig. 8h), thus excluding the possibility that Kap114p acts similarly to Mot1 upon recruitment by yTBP-DNA complexes[21].

**Fig. 5 | Kap114p functions as a negative regulator of yTBP and thus is part of the transcription regulatory network. a** A representative spot assay result showing growth of wild type (WT), single or double gene knockout strains of *KAP114* (Δ*kap114*) and *TAF1-TAND* (Δ*taf1-tand*) serially diluted (1:10) on minimal medium plates in the presence of 1.5 M NaCl and incubated at 30 or 37 °C. The assays were performed in triplicate. **b, c** WT, single or double gene knockout strains of *KAP114* (Δ*kap114*) and *TAF1-TAND* (Δ*taf1-tand*) were grown in minimal media in the presence of 1.5 M NaCl and incubated at 30 (**b**) or 37 (**c**)°C. Cell density at $OD_{600}$ nm was measured every day. Data are represented as mean ± SD ($n = 3$ independent experiments). Differences were assessed statistically by two-tailed Student's *t* test; n.s.: not significant; **$p < 0.01$; ***$p < 0.001$. **d** Schematic diagram of how Kap114p

and TAF1 antagonistically regulate TBP binding to promoter DNA. **e, f** A box-whisker plot showing fold-change of TBP ChIP-seq signals, relative to input DNA, at the promoters of mRNA ($n = 2284$) (**e**) or tRNA ($n = 272$) (**f**) genes for *KAP114* knockout strains rescued by vector control (-), wild-type *KAP114* or *KAP114*Δ(899-956) grown in the absence or presence of 1.5 M NaCl from a representative ChIP-seq experiment. A Kruskal–Wallis test with Dunn´s post-hoc test was used to determine the significance of each group. The *p* value from the Kruskal–Wallis test for each group is shown. Significant pairwise comparisons are annotated by an asterisk. ****$p < 0.0001$. Box limits: 25–75% quantiles, middle: median, upper (lower) whisker to the largest (smallest) value.

## The Kap114p pathway prevents yTBP dynamic collapse, conferring salt tolerance on yeast

Under high salt conditions, expression of *KAP114* in *KAP114*-knockout strains further suppressed occupancy of yTBP at tRNA promoter regions compared to under regular conditions (Fig. 5f and Supplementary Fig. 7b). However, yeast strains lacking *KAP114* did not exhibit increased binding of yTBP to promoters under high salt stress (Fig. 5f and Supplementary Fig. 7b), indicating that additional salt-dependent yTBP sequestration pathways are activated, perhaps leading to a collapse of the yTBP dynamic network and thereby accounting for yeast growth defects in high salt conditions. To test that hypothesis, we ectopically expressed *MOT1* and *TAF1* that negatively and positively regulate transcription, respectively, in the *KAP114*-knockout yeast strains[17,24]. In the *KAP114* knockout background, overexpression of *MOT1*, but not *TAF1*, resulted in a much stronger growth defect compared to *KAP114* single knockout, as demonstrated by our yeast spot-based assays and cell growth curves (Fig. 6a–e and Supplementary Fig. 8i). Notably, these results were only observed under salt stress but not under regular conditions (Fig. 6a–c and Supplementary Fig. 8i). Moreover, overexpression of *MOT1* in the WT strain did not elicit strong growth defects (Fig. 6a, d; lane 5). Mot1 is a TBP regulatory protein that sequesters TBP globally, thereby preventing TBP from binding DNA[21]. Hence, our results indicate that the Kap114p-dependent pathway competes with other TBP sequestration pathways to prevent yTBP dynamic collapse, thus conferring salt tolerance on yeast (Fig. 6f).

## Discussion

The karyopherin family comprises both karyopherins-α and -β. Although *Saccharomyces cerevisiae* only possesses one karyopherin-α gene, a few isoforms of karyopherin-α are found in humans[33]. Nevertheless, at least 14 and 24 Kap-βs have been identified in budding yeast and humans, respectively[34,35]. Karyopherins, together with the NPCs assembled by nucleoporins, RanGTPase, and the signal peptides in cargoes (e.g., NLS), are the major components that comprise the nuclear transport system. Karyopherins are generally considered to facilitate the passage of cargos across the central channel of NPCs. In the nuclear import pathway, many transcription factors are delivered by karyopherins to the nucleus (e.g., TBP and Sterol regulatory element-binding protein 2 (SREBP-2))[14,15,36]. Subsequently, RanGTP dissociates karyopherin-transcription factor complexes in the nucleus, so that transcription factors can exert their functions. To facilitate the transport of macromolecules across the NE, all components of the nuclear transport system must cooperate fully.

Notably, although 10 Kap-βs have been identified in budding yeast that function specifically as import receptors, only two of these are essential (i.e., Kap95p and Kap121p)[35]. Various combinations of the remaining "non-essential" Kap-βs can be deleted without apparent physiological consequences[15,37]. Moreover, multiple members of the Kap-β family have also been reported to "co-import" a single cargo, such as histone and TBP, to the nucleus[35]. Thus, several lines of evidence indicate functional redundancy of Kap-βs under regular conditions[35]. Many of the non-essential Kap-βs are evolutionarily

conserved from yeast to human[35], implying that their cellular functions beyond nuclear transport are most likely indispensable and retained throughout evolution.

Karyopherins do indeed conduct a variety of cellular processes in addition to nuclear transport under different circumstances. First, binding of karyopherins to the NLS of spindle assembly factors (SAFs) was proposed to negatively modulate mitotic spindle assembly[38]. Second, interaction of protein NLS with karyopherins positively promotes mitotic Golgi disassembly[39]. Hence, karyopherins coordinate two important events during mitosis (i.e., Golgi disassembly and spindle assembly), ensuring mitotic progression[39]. Third, karyopherins function as cytoplasmic chaperones to prevent protein aggregation of cargos with exposed basic domains[40]. Here, we have shown that, in addition to mediating nuclear import, Kap114p negatively regulates TBP and thus acts in the transcription regulatory network. Multiple lines of evidence support this scenario. First, Kap114p displays a high binding affinity for TBP (in the nanomolar range). Interestingly, Mot1p and TAF1 also interact with TBP with a high affinity (~1 nM), as revealed by multiple approaches[41–43]. Second, the Δ*taf1-tand* growth defect masked the impact of loss of *KAP114*, suggesting that *TAF1-TAND* is epistatic to *KAP114* and so *TAF1-TAND* functions downstream of *KAP114* under high salt conditions. Importantly, these results indicate that *Kap114* and *TAF1* function in the same pathway that regulates TBP activity. Hence, although the high binding affinity between Kap114p and TBP is energetically unfavorable for RanGTP-dependent cargo dissociation[16], it allows Kap114p to compete with different TBP interacting partners in the TBP regulatory network. Third, our cryo-EM structure reveals that Kap114p interacts with yTBP$^C$ using a binding mode structurally resembling that of other TBP regulatory proteins (e.g., TAF1 and Mot1). Fourth, biochemical analyses have further validated common motifs and residues shared by Kap114p and TBP regulatory proteins that mediate TBP interaction. Finally, fifth, our ChIP-seq analysis shows that Kap114p suppresses binding of yTBP to promoters.

The spring-like superhelical Kap-βs attributable to the array of linked α-helices display a large degree of flexibility, facilitating recognition of diverse cargos of various shapes and sizes[44]. Moreover, no conserved NLS sequences have been identified in yTBP and H2A-H2B in complex with Kap114p and Importin-9, respectively, indicating that their structural context rather than NLS sequence contributes to protein-protein interaction selectivity. We further propose that the structural flexibility provided by Kap-βs, together with structural contexts of their cargos, determines binding selectivity and affinity. Different binding modes and affinities for a single cargo could exist for a given Kap-β, enabling different cellular functions to be exerted. Biochemical and structural aspects of the high-affinity binding mode revealed herein for the Kap114p•yTBP$^C$ interaction highlight marked differences to that of the Importin-9-H2A-H2B complex[25]. Unexpectedly, the binding mode of the former is common to TBP regulatory proteins, implying convergent evolution of these two groups of proteins and supporting the proposal that Kap114p exerts a previously unappreciated function in transcriptional regulation.

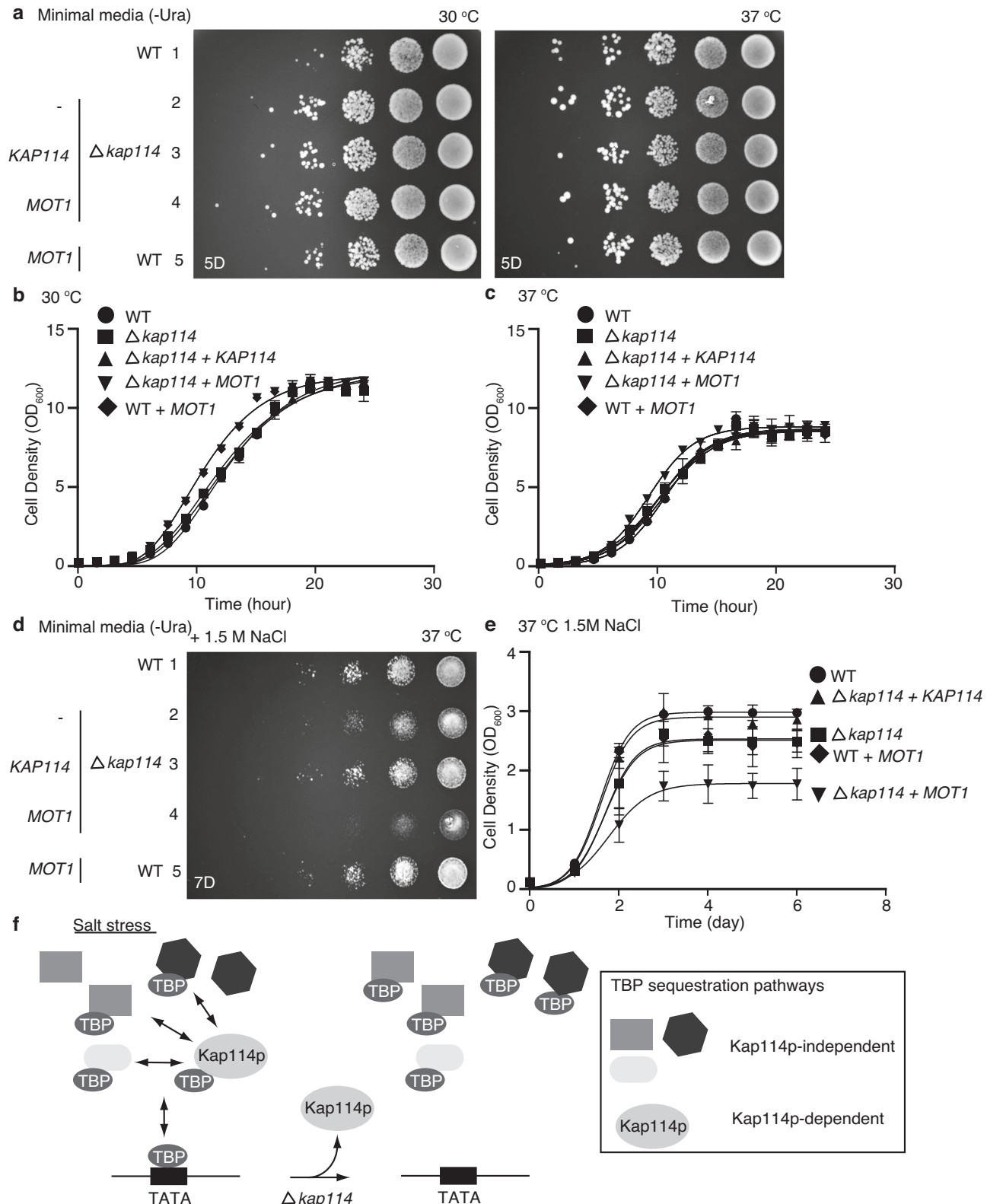

Notably, our biochemical analysis reveals that double point mutations of residues in Kap114p•yTBP$^C$ interacting surfaces (HEAT8B + loop and HEAT13B) greatly diminished binding between the Kap114p and yTBP$^C$ components (Fig. 4e, i). However, the Kap114p deletion mutant lacking the α-helix of the HEAT19 loop (a.a. 932-941) presented a binding affinity with yTBP$^C$ of ~150 nM (Fig. 2b). These results support that HEAT8B + loop and HEAT13B play critical roles in mediating yTBP$^C$

binding relative to the HEAT19 loop. The HEAT19 loop only partially contributes to the Kap114p and TBP interaction. However, our Chip-seq results have revealed that expression of wild-type *KAP114*, but not a HEAT19 loop deletion mutant, resulted in a statistically significant reduction of yTBP binding at the promoter DNA (Fig. 5e, f and Supplementary Fig. 7a, b), indicating that suppression of yTBP-promoter interaction by Kap114p involves its HEAT19 loop. Together with the

**Fig. 6 | A Kap114p-dependent pathway regulates the TBP network under high salt conditions. a** A representative spot assay result showing wild type and *KAP114* knockout strains expressing vector control (-), wild-type *KAP114* or *MOT1* and that were diluted serially (1:10), spotted onto minimal medium plates and incubated at 30 or 37 °C. The assays were performed in triplicate. **b, c** Wild type and *KAP114* knockout strains expressing vector control (-), wild-type *KAP114* or *MOT1* were grown in minimal media in the presence of 1.5 M NaCl and incubated at 30 (**b**) or 37 (**c**) °C. Cell density at $OD_{600}$ nm was measured every 90 min. Data are presented as mean ± SD (*n* = 3 biologically independent experiments). **d** A representative spot assay result showing wild type and *KAP114* knockout strains expressing vector control (-), wild-type *KAP114* or *MOT1* and that were diluted serially (1:5), spotted onto minimal medium plates in the presence of 1.5 M NaCl and incubated at 37 °C. The assays were performed in triplicate. **e** Wild type and *KAP114* knockout strains expressing vector control (-), wild-type *KAP114* or *MOT1* were grown in minimal media in the presence of 1.5 M NaCl and incubated at 37 °C. Cell density at $OD_{600}$ nm was measured every day. Data are presented as mean ± SD (*n* = 3 biologically independent experiments). **f** Schematic illustrating that a Kap114p-dependent pathway competes with other TBP sequestration pathways to prevent yTBP dynamic collapse.

finding that the HEAT19 loop deletion mutant can rescue the mislocalization of yTBP elicited by *KAP114* knockout[16], we propose that the HEAT19 loop of Kap114p modulates TBP-mediated transcription but is unlikely to be essential for the nuclear import of TBP. Likewise, the latch deletion mutant of Mot1 still formed a complex with TBP, suggesting that it also minimally contributes to TBP binding[21]. Instead of facilitating Mot1-TBP interaction, the latch region of Mot1 renders the TBP–DNA complex less stable, enhancing displacement of TBP from DNA.

Notably, we did not observe a large conformational rearrangement in the Kap114p solenoid upon Kap114p•yTBP$^C$ complex formation (Supplementary Fig 3e, f). The positive ΔS we determined supports an entropically-driven endothermic reaction for Kap114p and yTBP$^C$ assembly (Supplementary Table 1). The endothermic reaction shown in the ITC data may be attributed to the existence of protein-solvent interactions (e.g., water molecules), even though no water molecules could be assigned in our low-resolution cryo-EM structure. Energy is needed to disrupt protein-solvent interactions (e.g., those of Kap114p and yTBP$^C$ with water) so that hydrogen bonds and salt bridges may form in the complex (protein-protein interactions) to facilitate Kap114p and yTBP$^C$ binding. These two opposing forces have been proposed as being a general feature of highly charged complex assembly[45]. In this system, it is most likely that more energy is expended breaking "pre-existing" bonds (protein-solvent) than is released to form bonds (protein-protein). Alternatively, the large conformational change in the HEAT19 loop upon Kap114p binding to yTBP contributes to the endothermic reaction, since significant domain movement upon substrate binding has also been demonstrated to contribute to an entropy-driven process[46].

In budding yeast, exposure of cells to different stress conditions (e.g., oxidative and ethanol) promotes post-translational modification of proteins by SUMOylation[47]. Moreover, conjugation of the ubiquitin-like protein SUMO to substrates is enhanced in plants under salt stress[48]. Notably, Kap114p can be SUMOylated at lysine 909, which modulates binding of Kap114p to its cargos[49]. Residue K909 lies within the unstructured region of the HEAT19 loop according to our cryo-EM structure of the Kap114p-yTBP complex. Moreover, the Ran gradient is perturbed under stress conditions, such as under high salt[16,50]. Thus, it is possible that Kap114p SUMOylation and the Ran gradient modulate the Kap114p and yTBP interaction, providing additional levels of regulation under salt stress.

## Methods
### Cloning procedures
*S. cerevisiae* full-length Kap95p, Kap114p from total genomic DNA, and Kap114p mutants [Kap114pΔ(H18loop), residues 899–956 replaced with CCCGGG linker; Kap114pΔ(932-941); Kap114pΔ(Y933); Kap114pΔ(Y939); Kap114p(Y370A); Kap114p(Q597A)] were amplified by PCR and cloned into the modified pGEX6p1 expression vector (GE Healthcare) that contains a GST tag followed by a PreScission cleavage site. *S. cerevisiae* wild-type yTBP$^C$ and yTBP$^C$ mutants [yTBP(N91A); yTBP(R141A); yTBP(K145A)] were cloned into a modified pET28a vector containing a 6xHis tag followed by a PreScission cleavage site at the N-terminus. All constructs are listed in Supplementary Table 4.

## Protein expression and purification
To purify recombinant Kap114p proteins, individual proteins were expressed and purified. Transformed *Escherichia coli* cells (Rosetta strain) were grown in LB medium at 37 °C until $OD_{600nm}$ reached 0.6-0.8. The temperature was then lowered to 18 °C and cells were induced with 0.5 mM IPTG. After overnight induction, cells were collected by centrifugation (5000 *g* for 10 min at 4 °C). Pelleted cells that contained GST-tagged proteins were resuspended in lysis buffer (20 mM HEPES pH 7.4, 150 mM NaCl, 3 mM DTT). After being lysed with a French press, the disrupted cells were centrifuged for 30 min at 34,000 *g*, 4 °C. The cleared supernatant was mixed and incubated for 40 min at 4 °C with GST resin (GE Healthcare) pre-equilibrated in lysis buffer. The resin with bound GST-tagged protein was washed thoroughly in batches with the lysis buffer and then transferred to a packing column (GE Healthcare). The column was washed with lysis buffer and the protein was eluted with elution buffer (20 mM HEPES pH 7.4, 150 mM NaCl, 3 mM DTT, 50 mM glutathione). Eluted protein was collected and dialyzed against buffer (20 mM HEPES pH 7.4, 50 mM NaCl, 3 mM DTT) overnight at 4 °C (PreScission protease was added during dialysis for GST-tag cleavage). After dialysis, the protein was diluted to a salt concentration of 50 mM and then loaded onto a MonoQ 5/50 GL column (GE Healthcare). The fractions were collected by eluting with elution buffer (20 mM HEPES pH 7.4, 500 mM NaCl, 3 mM DTT). The sample was then loaded onto a size-exclusion chromatography Superdex 200 Increase 16/60 column (GE Healthcare) in a running buffer containing 20 mM HEPES pH 7.4, 50 mM NaCl, and 3 mM DTT, before being analyzed by SDS-PAGE. Eluted protein fractions were collected, concentrated, and aliquoted into PCR tubes. Aliquoted vials were rapidly frozen in liquid nitrogen and stored in a −80 °C freezer for further experimental use.

For yTBP$^C$ protein expression, genes encoding wild type and mutant variants were transformed into *E. coli* cells (Rosetta strain) and grown in LB medium at 37 °C until $OD_{600nm}$ reached 0.6–0.8. The temperature was then lowered to 18 °C and cells were induced with 0.5 mM IPTG. After overnight induction, cells were collected by centrifugation (5000 *g* for 10 min at 4 °C). Pelleted cells that contained His-tagged proteins were resuspended in lysis buffer (50 mM $KH_2PO_4$, 50 mM $Na_2HPO_4$, 300 mM NaCl, 3 mM β-mercaptoethanol, pH 7.4). After being lysed with a French press, the disrupted cells were centrifuged for 30 min at 34,000 *g*, 4 °C. The cleared supernatant was mixed and incubated for 40 min at 4 °C with Ni-NTA beads (QIAGEN) pre-equilibrated in lysis buffer. The resin with bound His-tagged protein was washed thoroughly in batches with the lysis buffer and then transferred to a packing column (GE Healthcare). The protein was first prewashed with lysis buffer (50 mM $KH_2PO_4$, 50 mM $Na_2HPO_4$, 300 mM NaCl, 3 mM β-mercaptoethanol, pH 7.4), followed by 5% elution buffer wash (25 mM imidazole), and then purified by affinity chromatography using 50% elution buffer (250 mM imidazole). Eluted protein was collected and dialyzed against buffer (20 mM HEPES pH 7.4, 300 mM NaCl, 3 mM DTT) overnight at 4 °C (PreScission protease was added during dialysis for His-tag cleavage). After dialysis, the protein then underwent ion-exchange chromatography by loading the sample into a HiTrap SP HP (5 ml) column with running buffer (20 mM HEPES pH 7.4, 300 mM NaCl, 3 mM DTT). The fractions were collected

by eluting with elution buffer (20 mM HEPES pH 7.4, 500 mM NaCl, 3 mM DTT). The sample was then loaded onto a size-exclusion chromatography Superdex 75 Increase 16/60 column (GE Healthcare) in a running buffer. Eluted protein fractions were analyzed by SDS-PAGE, and then concentrated and aliquoted into PCR tubes. Aliquoted vials were rapidly frozen in liquid nitrogen and stored in a -80 °C freezer for further experimental use.

### Complex assembly and cross-linking
Purified Kap114p•yTBP$^C$ complex (-28 mg/mL) was crosslinked by adding glutaraldehyde to a final concentration of 0.01% and the reaction solution was incubated for 10 min at room temperature. The reaction was stopped by adding 1 M Tris/HCl pH 7.5 (100 mM final concentration). To remove any higher-order cross-links and aggregates, the sample was further purified using a size-exclusion Superdex 200 Increase 10/30 column. Cross-links were checked by running eluted fractions on 15 % SDS-PAGE and collected peak fractions were determined according to the gel filtration profiles.

### Electron microscopy sample preparation
To prepare cryo-EM grids, the cross-linked Kap114p•yTBP$^C$ complex was diluted to - 0.7 mg/mL. Four microliters of the sample were loaded onto a glow-discharged Holey carbon (Quantifoil R2/1) grid (Quantifoil Micro Tools GmbH). After blotting (3 sec), grids were plunge-frozen in liquid ethane using an FEI Vitrobot system (Thermo Fisher Scientific). The temperature in the chamber was kept at 4 °C and 100% humidity.

### Electron microscopy data collection
Cryo-EM grids were first checked on a Talos transmission electron microscope equipped with a Falcon III detector (Thermo Fisher Scientific) operated in linear mode. The images were recorded at a nominal magnification of 120,000x, corresponding to a pixel size of 0.86 Å/pixel, with a defocus setting of -3.0 μm. Suitable cryo-EM grids were recovered for further data collection on a 300 kV Titan Krios transmission electron microscope hosting a K3 detector (with GIF Bio-Quantum Energy Filters, Gatan) operating in super-resolution mode and using EPU-2.7.0 software (Thermo Fisher Scientific). The raw movie stacks were recorded at a magnification of 105,000×, corresponding to a pixel size of 0.83 Å/pixel (super-resolution 0.415 Å/pixel). The defocus range was set to −1.5 to -2.5 μm and the slit width of energy filters was set to 20 eV. Forty frames of non-gain-normalized tiff stacks were recorded with a dose rate of -16 e-/Å2 per second and the total exposure time was set to 2.5 s, resulting in an accumulated dose of ~40 e-/Å2 (-1.0 e-/Å2 per frame). The parameters for cryo-EM data acquisition are summarized in Table 1.

### Single-particle image processing and 3D reconstruction
The super-resolution image stacks were imported into Relion (Zivanov et al., 2018) for motion-correction and dose-weighting using Motion-Cor2 (Zheng et al., 2017), with a 5 × 5 patch and two-fold binning (resulting in a pixel size of 0.83 Å/pixel). The motion-corrected micrographs were then imported into cryoSPARC[51] for further single-particle reconstruction. The contrast transfer function (CTF) was determined from the motion-corrected images using the "Patch CTF estimation" function in cryoSPARC. Selected particles were extracted with a box size of 256 pixels for further 2D classification. Poor 2D class averages were removed after 2D classification and the remaining particles were used for ab initio map generation, followed by 3D heterogeneous refinement (separated into four classes). Poor 3D classes were removed via several rounds of 3D heterogeneous refinement, and particles in good 3D classes were merged before conducting an additional 2D classification to remove any remaining bad particles. The good particles were further refined by 3D homogeneous refinement and non-uniform refinement without imposed symmetry. Map sharpening and resolution estimation were automatically performed via 3D

non-uniform refinement in cryoSPARC (Punjani et al., 2017). The overall resolution was estimated using the Fourier Shell Correlation (FSC) = 0.143 criterion and the local resolution was also calculated in cryoSPARC (Punjani et al., 2017). The 3D density maps were visualized in UCSF Chimera (Yang et al., 2012). The procedures for single-particle image processing and details of cryo-EM reconstruction are summarized in Supplementary Fig. 2. Statistical information for cryo-EM reconstructions are summarized in Table 1.

### Isothermal titration calorimetry (ITC)
Wild type and mutant Kap114p and wild type and mutant yTBP$^C$ were expressed and purified as described above. The purified proteins were diluted in ITC buffer (20 mM HEPES pH 7.4, 150 mM NaCl). ITC experiments were carried out using an ITC-200 calorimeter (MicroCal iTC200) at 25 °C with 25 or 50 μM of wild type or mutant Kap114p proteins in the sample cell and 250 or 500 μM wild-type or mutant yTBP$^C$ proteins in the syringe. Twenty injections (4 μl sample for each) were sequentially made in each experiment (note, the first injection was 2 μl). The injections were mixed at 750 rpm, with an interval of 180 sec between injections to allow the peak to return to baseline. All experiments were carried out in triplicate. The experimental data were fitted to theoretical titration curves using MicroCal software (ORIGIN). Individual experiments in the triplicate sets are differentially color-coded. Buffer controls are shown in Supplementary Fig. 4b, c. No heat change was observed in the reference experiments, so the final data presented has not been reference-subtracted.

### Yeast strain construction
*MOT1, TAF1*, or *TAF1-TAND* (including the upstream 500 base pairs (bp)) was cloned into the vector pRS426. Each of the PCR products was transformed into YTK12029 (a gift from Dr. T. Kokubo) and *KAP114* knockout yeast strains generated previously. The *KAP114* knockout strain SC1005 was transformed with pRS426 as a control or pRS426 carrying wild-type *MOT1, TAF1*, or *TAF1-TAND*. To generate the YTK12029 strain carrying GFP-labeled *KAP114* and *KAP114Δ(899–956)*, the yeast strain was transformed with pRS426 carrying GFP-labeled *KAP114* and *KAP114Δ(899–956)*. All yeast strains used in this study are listed in Supplementary Table 3.

### Yeast spot-based assay
Yeast strain YTK12029 and the *KAP114* knockout strain were transformed with pRS426 as a control or pRS426 carrying wild-type *MOT1, TAF1*, or *TAF1-TAND*. Control and over-expressed yeast strains were grown at 30 °C in a minimal medium (-Ura). For dilution spot assays, yeast samples grown to logarithmic phase were initially diluted to the same OD. Subsequently, a series of ten-fold dilutions was generated, and then 10 μl of each sample was spotted onto a selective medium and incubated at the indicated temperatures. To examine yeast growth under stress conditions, yeast spot assays were performed using YPD medium to which was added 1.5 M NaCl and incubated at the indicated temperatures.

### RT-qPCR
Yeast strains (BY4741, YER148W-GFP, SC1005, and SC1007) expressing vector control, *KAP114*, or *KAP114Δ(899–956)* were grown in minimal medium (-Ura) with or without 1.5 M NaCl at 30 °C to reach logarithmic phase ($OD_{600nm}$ = 1.0). Total RNAs were extracted using a RNeasy Mini Kit (QIAGEN) and were reverse-transcribed to cDNA using RevertAid Reverse Transcriptase (ThermoScientific) in a 20 μL reaction system. Quantitative PCR (qPCR) was performed using Fast SYBR™ Green Master Mix (ThermoScientific) in a QuantStudio™ 12 K Flex Real-Time PCR System (ThermoScientific). RT-qPCR data were analyzed by the comparative Ct method (ΔΔCt) and relative expression levels were normalized to the level of Actin. Primer sequences of each gene tested are listed in Supplementary Table 2.

## Immunoblotting

Yeast SC1007 strains expressing vector control, *KAP114* or *KAP114Δ(899–956)* were grown in minimal medium (-Ura) with or without 1.5 M NaCl at 30 °C to reach logarithmic phase (OD$_{600nm}$ = 1.0). The yeast cell lysates were extracted by mechanically shaking cells mixed with the Glass Beads (BioSpec; Cat. No. 11079105; Mini-BeadBeater Glass Mill Beads, 0.5 mm dia.) in lysis buffer (50 mM Tris-HCl pH 7.5, 1 mM EDTA, 150 mM NaCl, 0.05% Triton X100, 10% glycerol, 1 mM dithiothreitol, 1 mM phenylmethylsulfonyl fluoride (PMSF)) and analyzed by 12% gradient SDS-PAGE. For the analysis of the expression levels of GFP-yTBP among the yeast strains examined, the cell lysates were transferred onto 0.2 μm Immobilon®-PSQ PVDF Membrane (MERCK) using the manufacturer's setting of the Mini Trans-Blot® Electrophoretic Transfer Cell (Bio-Rad). Subsequently, membranes were blocked in 3% milk in TBS-T (0.01% Tween-20) for 30 min at room temperature while shaking. Afterward, the membranes were incubated with the rabbit polyclonal anti-GAPDH antibody (GeneTex; GTX100118; Lot. No. 42977) and the rabbit monoclonal anti-GFP antibody (EPR14104; ab183734; Lot. No. GR298298-33), respectively. Primary antibodies were diluted at 1:2500 in 3% milk in TBS-T. The membranes were incubated with the primary antibody at 4 °C overnight. Afterward, membranes were washed three times (10 min at room temperature each time) with TBS-T prior to applying the secondary antibody. HRP-conjugated secondary goat anti-rabbit IgG antibody (GeneTex; GTX213110-01; Lot. No. 43262; Goat pAb) diluted 1:10,000 in 3% milk in TBS-T was incubated for 1 h at room temperature while shaking. Afterward, membranes were washed three times (10 min at room temperature each time) with TBS-T before the horseradish peroxidase (HRP) chemiluminescence kit (GeneTex) was used for detection. Band intensities were detected using a BioSpectrum 815 CCD camera and analyzed in VisionWork software.

## Cell density measurement

Yeast strains were grown overnight in a minimal medium with or without 1.5 M NaCl at 30 °C or 37 °C. The OD$_{600}$ values of 20-fold diluted cultures were monitored using a photoelectric colorimeter (Eppendorf BioPhotometer D30). Three independent experiments were carried out.

## ChIP-qPCR, ChIP-seq, and data analysis

Yeast strain SC1007 expressing vector control, *KAP114* or *KAP114Δ(899–956)* were grown in minimal medium (-Ura) with or without 1.5 M NaCl at 30 °C to reach logarithmic phase (OD$_{600nm}$ = 0.8). Cells were cross-linked with 1% formaldehyde (Sigma) for 15 min and subsequently quenched with 125 mM glycine for 5 min at room temperature. Fixed cells were washed with ice-cold ST buffer (10 mM Tris–HCl, pH 7.5, 100 mM NaCl) and lysed in FA lysis buffer (50 mM HEPES-KOH pH 7.5, 150 mM NaCl, 2 mM EDTA, 1% Triton X-100, 0.1% sodium deoxycholate, and 1X Roche protease inhibitor cocktail) with a bead beater (MP Biomedicals). The crosslinked chromatins were pelleted by centrifugation and fragmented in FA lysis buffer supplemented with 0.1% SDS using an ultrasonic homogenizer (Q125 Qsonica). For ChIP-qPCR assays, ChIP was performed using 50 mL culture-equivalent chromatin, 2.5 μg anti-GFP antibody (Abcam ab290) and 10 μL protein G Dynabeads (Invitrogen). The ChIPed DNA was quantified by PCR with SYBR green master mix in a StepOnePlusTM Real-Time PCR System (Applied Biosystems). Normalized Ct (ΔΔCt) values were calculated by subtracting the Ct value of the promoter of indicated genes from the Ct value of a reference region (1 kb downstream of the *ACT* promoter). Data are presented as fold enrichment relative to the aforementioned reference region. Gene-specific primers are listed in Supplementary Table 2. For ChIP-seq assays, 10 μg anti-GFP antibody and 25 μL protein G Dynabeads were conjugated in an antibody blocking buffer (PBS with 0.5% BSA and 0.05% TX100) for 2 h. Two hundred OD (~500 mL culture-equivalent) of yeast chromatin was

incubated with above anti-GFP antibody-bead conjugates overnight. After extensive washing with high salt buffer (20 mM HEPES-KOH pH 7.5, 500 mM NaCl, 0.1% SDS, 1% TX100, 2 mM EDTA), the immuno-precipitates were eluted using TE buffer supplemented with 1% SDS and then subjected to RNase A and Proteinase K digestion. Large DNA fragments were depleted by means of AMPure XP beads (Beckman Coulter). ChIPed DNA samples were further purified using a PCR purification kit (Qiagen) and concentrations were determined using a Quantus Fluorometer (Promega). Two to five ng DNA was sent to the IMB Genomics Core (Academia Sinica) for ChIP library construction and next-generation sequencing in a NextSeq500 system according to Illumina instructions. FastQC was used for quality control of raw sequencing data. High-throughput raw reads were aligned to *Saccharomyces cerevisiae* genome assembly R64 (sacCer3) using bowtie (v.1.1.2) with parameters "-n 2 -m 1 -l 36 --best". The number of total reads, uniquely aligned reads, and mapping efficiency are listed in Supplementary Table 5. PCR duplicates were removed using SAMtools (v1.9). Tag density at promoters (TSS ± 100 bp) of *Saccharomyces* Genome Database (SDG)-annotated tRNA and mRNA genes was determined using the *analyzeRepeats* command of HOMER software and the results are presented in Source Data. To compare TBP occupancy, mRNA genes with higher TBP ChIP-seq signals [tag density > 100 rpkm and 2-fold (Figs. 5e, f) or 3-fold (Supplementary Fig. 7a, b) enrichment over input DNA] in *KAP114* knockout cells were used. For tRNA genes, we compared 272 out of 275 SDG-annotated tRNA genes (the remaining three genes showed no tag density in either input DNA or ChIP-seq samples).

## Data availability

All experimental data used in this study are available within the main text, Supplementary Information file, and Source Data file. The source data underlying Figs. 2a-e, 4c-k, 5a-c, 5e, 5f, 6a-c, 6d-e and Supplementary Figs. 4a-g, 6, 7a-b, and 8 are provided in Source Data file. The ChIP-seq data generated in this study have been deposited in the NCBI/Gene Expression Omnibus (GEO) database under accession code no. GSE217150. Protein structural coordinates and maps have been deposited in the PDB (8H5B) and the Electron Microscopy Data Bank (EMD) (EMD-34490). The accession codes of previously solved structures used in this study are Kap114p (6AHO), yTBPc (1YTB), Importin-9•H2A-H2B (6N1Z), Importin-β•Ran (2BKU), Mot1-NTD•*Ec*TBP (3OC3), and yTAF-TAND•yTBP (4B0A). The validation reports of the corresponding coordinate files are provided. Source data are provided with this paper.

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

## Acknowledgements

The cryo-EM experiments were performed at the Academia Sinica Cryo-EM Facility (ASCEM) and Academia Sinica Grid-computing Center (ASGC). ASCEM is jointly supported by Academia Sinica Core Facility and Innovative Instrument Project Grant No. AS-CFII-111-210 and Taiwan Protein Project Grant No. AS-KPQ-109-TPP2. We acknowledge the use of the ITC system in the Biophysics Core Facility, funded by Academia Sinica Core Facility and Innovative Instrument Project AS-CFII1-111-201. We also thank John O'Brien for manuscript editing, S.-Y. Tung from the IMB Genomics Core facility, and H.-N. Lin, C.-H. Yu, and K.-H. Yeh from the IMB Bioinformatics Core facility for technical assistance. We are grateful to T.L. Lowary and C. Lim for comments and suggestions on the manuscript. This work was supported by Academia Sinica (AS-IVA-112-L05 to K.-C.H.), National Science and Technology Council (NSTC-109-2311-B-001-020-MY3 to K.-C.H.; NSTC 110-2320-B-A49A-533-MY3 to W.-Y.C.), Cancer Progression Research Center (113W31101), Cancer and Immunology Research Center (112W31101), National Yang Ming Chiao Tung University, and Featured Areas Research Center Program within the framework of the Higher Education Sprout Project by the Ministry of Education in Taiwan (to W.-Y.C.).

## Author contributions

Author contributions: C.-C.L. and Y.-S.W. performed biochemical experiments. C.-C.L. conducted qPCR assays. C.-C.L., C.-H.W., and Y.-M.W. determined the cryo-EM structure. W.-C.P. and W.-Y.C. carried out the Chip-seq analysis. W.-Y.C. and K.-C.H. prepared the manuscript with input from all authors. All authors discussed the results and commends on the manuscript. K.-C.H. supervised the overall work.

## Competing interests

The authors declare no competing interests.
