## [Peer review file · Nature Communications]

REVIEWER COMMENTS

Reviewer #1 (Remarks to the Author):

TATA box-binding protein (TBP) is a general transcription factor that binds AT-rich sequences at core promoters and is involved in transcription initiation by all three RNA polymerases. The protein contains a conserved 180 amino acids C-terminal domain made up of two pseudo-symmetrical repeats, that mediates the interaction with both DNA and diverse transcription regulators, exploiting different binding surfaces (concave and convex, respectively).

To fulfill its functions, TBP must be imported within nuclei across the nuclear envelope. In yeast this process is mediated by a group of nuclear transport receptors named karyopherin- β (Kap- β). Yeast TBP (γ TBP) interacts with many Kap- β s showing a remarkable redundancy in their nuclear import. Some Kap- β s seem not to be essential for yeast viability, therefore their interaction with γ TBP may play a critical role for the modulation of gene expression.

The major import pathway of γ TBP into the nucleus is mediated by the Kap family member, Kap114p, which is not an essential protein [Lucy F. Pemberton et al., *The Journal of Cell Biology*, 145:7, 1999, p:1407–1417].

The authors, previously demonstrated that γ TBP interacts with Kap114p with a nM affinity, almost four orders of magnitude higher than for other Kap- β s. [Chung-Chi Liao et al., *EMBO Rep.* 21:2, e48324, 2020]. However, the molecular details of this interaction are still missing.

This manuscript reports a thorough molecular and structural characterization of the interaction between γ TBP and Kap114p. The structural results obtained by implementing single-particle cryo-EM illustrate the determinants of the interaction between γ TBP and the C-Terminal domain of Kap114p, revealing the critical residues involved in the binding. The interaction is further corroborated by other mutagenesis and Isothermal titration calorimetry (ITC) studies. Moreover biochemical analysis allowed to speculate that Kap114p dependent regulation pathways compete with other TBP sequestration pathways to confer salt tolerance to the yeast.

The results of this work have important implications for our understanding of the modulation of gene expression within nuclei and significantly contribute to improve our insight into the transcriptional regulatory network.

In general, the article is well written and I support the publication of this manuscript in *Nature Communication* after addressing the critical points listed below.

Minor points:

- For more clarity, I suggest the authors to indicate also the HEAT8 and HEAT13B loops in figure 1B, perhaps choosing different colors.
- In some figures (e.g., Fig. 2b) the text is overlapping with the data making it difficult to comprehend.
- Please notice that some typos are present throughout the text. For example, in the ITC section page 25 line 22 you write “50mM of wild type or mutant Kap114p” and “500 mM of wild type or mutant γ TBPc” but I think it should read 50 μ M and 500 μ M.
- The concentrations used for ITC experiments are much higher than the K_d found by the authors for this interaction, therefore the error is also quite significant. It would be beneficial, if possible, to repeat the titration at lower concentrations (e.g., 8 μ M (cell) and 80 μ M (syringe)).
- At page 12 line 13 the authors claim that “simultaneous depletion of both kap114 and TAF1-TAND partially rescued the defective growth phenotypes caused by sole TAF1-TAND disruption, implying functional antagonism between these two TBP-interacting factors”. However, looking at Fig. 5a, it seems to me that both kap114 and TAF1-TAND depleted mutants display growth defects at high salt concentrations and I would expect a much stronger effect for the mutant lacking both proteins, but this seems not to be the case. If possible, I would suggest the authors to provide further explanations within the text.

Major points:

- The “materials and methods” section of the ITC data does not mention if reference experiments were acquired and subtracted from final data. However, this is a critical step for any ITC experiments and it is fundamental to be able to reproduce the results presented in the manuscript. Therefore, it is important to include reference experiments in the manuscript.
- The ITC data presented are endothermic. This might be due to the presence of additional hydration. However, it seems that the complex is mainly characterized by the formation of hydrogen bonds and salt bridges, that usually are associated with exothermic processes. It will be useful to discuss a bit more in detail the ITC data on the base of the Cryo-EM structure and to include also the ΔH and ΔS values obtained using ITC in table 1. In particular, the authors describe on page 9 line 4 that they did not observe any large conformational rearrangement upon complex formation, suggesting that there should be a favorable ΔS variation (positive ΔS) due to enhanced disorder in the environment. Including ΔS and ΔH data will be beneficial to further support and understand the structural data.

Reviewer #2 (Remarks to the Author):

Liao et al. report a high resolution cryoEM structure of Kap114p bound to cargo TATA-box binding protein to reveal the structural basis of the high affinity binding of the cargo, as well as using yeast assays and structural guided mutants to study its transcriptional repressor function. Overall, the study is performed carefully, and the manuscript is written well. However, I do have a few comments that I believe will help improve the manuscript for publication.

Major concerns:

1. The Heat 19 loop is shown to be critical for binding and repressor function. Since its interaction is shown in the main figure, it would be good to show the density for this peptide on the map in supplementary so that readers can assess the confidence in the placement of the helix. It is hard to tell the density in extended data Figure 3A.
2. While the interaction interfaces are all investigated by mutagenesis, I feel that discussion is missing for the comparison of the interfaces and their contributions to affinity and function. Based on ITC mutagenesis, it seems like Heat8 interaction seems to be a lot more important than the HEAT 19 loop since the affinity dropped to micromolar range. The HEAT 19 loop deletion only drops the affinity to hundred of nanomolar. I wonder whether H19 loop deletion is still viable for import but the drop of affinity is sufficient to affect the transcriptional repression activity.
3. Since Kap114 is being compared to TAF1 and MOT1 as having similar interactions to γ TBP, it would be informative to compare the binding affinities - either ITC measurement, or citing previous literature if available. This will also tie in with my last concern.

Minor concern:

1. Q597 can not be seen in Figure 4, but is a critical mutant that was used to validate modeling, perhaps another angle of the structure can be chosen to indicate the position of the residue.

Reviewer #3 (Remarks to the Author):

The manuscript by Liao et al. reports the structure of the TATA-box binding protein (TBP) in interaction with the karyopherin Kap114. The authors also provide functional evidence that Kap114 might contribute to TBP promoter recruitment and regulatory roles in *Saccharomyces cerevisiae*. This study builds up on previous work by the same group that identified a role of Kap114 in the nuclear import of TBP. Here the authors used cryo-electron microscopy complemented with biochemical assays to thoroughly characterize Kap114-TBP interaction, revealing that Kap114 engages with TBP in similar manner as other important TBP regulators, including the TFIID subunit TAF1 and the Mot1 ATPase. Proliferation assays, genetic analyses and ChIP-seq analyses were performed to examine the effect of Kap114 on TBP function in vivo, but these experiments suffer from several flaws and are

overinterpreted, making this work too preliminary for publication in its current form.

Specifically, I have the following concerns with the data presented in Figure 5 and Extended Figure 5 and how they are described in the text.

First there are major issues with the interpretation of the drop tests (Fig. 5a and Extended Fig. 5d).

- The conclusion that the Taf1-TAND antagonizes Kap114 for TBP function because the TAND deletion would rescue the loss of Kap114 is not supported by the data shown in Fig. 5a. The spot assays show, rather, that the Taf1-TAND and Kap114 regulate high salt tolerance independently because the double mutant displays intermediate growth as compared to each single mutant. In contrast, at 37C, the Taf1-TAND growth defect is epistatic to the loss of Kap114, suggesting that the Taf1-TAND functions downstream of Kap114 in this condition.

- Kap114 mutants that specifically abolishes its interaction with TBP are not tested, although the authors identified a specific region and the residues that are involved. Given the probable pleiotropy of kap114 mutant phenotypes, it is not possible to establish a direct link between the salt/heat stress sensitivity and the TBP binding role of Kap114 with a full kap114 deletion mutant.

- Fig. 5a shows that the kap114 deletion mutant has a growth defect at 37C, thus under chronic heat stress, rather than osmotic stress, since the mutant strain grows as well as the wild-type control at 30C in 1.5M NaCl. The mutant might be sensitive to a combination of both stresses, but this is not tested explicitly. For this, the authors need to show spot assays performed in all four conditions (no stress, 1.5M NaCl at 30C, 37C alone, and 1.5M NaCl at 37C).

- A similar growth control is lacking for the Mot1 overexpression tests, which are spread between Fig. 5e and Extended Fig. 5d (consider to merge these panels into one figure for clarity). In addition, the effect of Mot1 on Kap114 must be tested against a strain that overexpresses Mot1 alone, as a control, to determine whether the growth defects induced by Mot1 overexpression depend on the loss of Kap114.

- Finally, it is imperative to confirm these spot assays, which are semi-quantitative, with an orthogonal approach to measure yeast proliferation, such as measuring cell concentration during the logarithmic phase of growth in each condition, with independent biological replicates. There are actually no details about how many independent replicates were performed for each spot assay.

Second, there are important issues with the ChIP data. Specifically:

- The authors need to verify the levels of GFP-tagged TBP as well as of the WT and mutant Kap114 proteins using Western blotting, both in normal and high salt/heat stress conditions, to control for the putative differences in ChIP signal shown in Fig. 5c,d.

- The authors need to provide evidence that GFP-tagged TBP is functional, particularly because its binding is much lower than expected at Pol II-dependent promoters (only 1480 genes identified).

- The legend of Fig. 5c,d indicates that the data are represented as box-and-whisker plots, as seen on the graph. However, further down, the legend indicates that 'The results of three biological replicates are shown as mean \pm SD' - is this really what the whiskers are? There would tremendous variability then.

- Using a two-tailed Student's t test to analyze the ChIP-seq data on Fig. 5c,d is not possible for several reasons, most importantly because it does not correct for multiple testing and, if I understood correctly that each dot represents one TBP peak, because these peaks do not correspond to independent replicates. Furthermore, the reporting summary indicates that ChIP-seq were done only

once (1 replicate), precluding any differential analyses and interpretation of differences between strains and conditions.

Minor concerns:

Page 2 Lane 4: Pol I transcribes rRNAs and Pol III transcribes tRNAs so they should be swapped in that sentence. Using 'respectively' requires to respect the order in each group of words.

Page 9: I do not really understand what Fig 2f is supposed to show. The data presented in Ext. 4 describe clearly that Taf1, Mot1, and Kap114 employ both similar and distinct structural features to interact with TBP. This part should be described in greater details in the text.

Page 9-10: The sentence that starts at the end of page 9 and ends on page 10 is confusing. It is difficult to understand which partner the authors are referring to, Kap114 or TBP, when describing the residues. Fig. 2i does not mention which protein sequences are shown, further complicating this sentence.

Extended Fig. 5a lacks numbers for the y-axis and a schematic depiction of the locus shown, along with a scale bar for genomic distance.

Reviewer #1 (Remarks to the Author):

TATA box-binding protein (TBP) is a general transcription factor that binds AT-rich sequences at core promoters and is involved in transcription initiation by all three RNA polymerases. The protein contains a conserved 180 amino acids C-terminal domain made up of two pseudo-symmetrical repeats, that mediates the interaction with both DNA and diverse transcription regulators, exploiting different binding surfaces (concave and convex, respectively).

To fulfill its functions, TBP must be imported within nuclei across the nuclear envelope. In yeast this process is mediated by a group of nuclear transport receptors named karyopherin- β (Kap- β). Yeast TBP (yTBP) interacts with many Kap- β s showing a remarkable redundancy in their nuclear import. Some Kap- β s seem not to be essential for yeast viability, therefore their interaction with yTBP may play a critical role for the modulation of gene expression.

The major import pathway of yTBP into the nucleus is mediated by the Kap family member, Kap114p, which is not an essential protein [Lucy F. Pemberton et al., *The Journal of Cell Biology*, 145:7, 1999, p:1407–1417].

The authors, previously demonstrated that yTBP interacts with Kap114p with a nM affinity, almost four orders of magnitude higher than for other Kap- β s. [Chung-Chi Liao et al., *EMBO Rep.* 21:2, e48324, 2020]. However, the molecular details of this interaction are still missing.

This manuscript reports a thorough molecular and structural characterization of the interaction between yTBP and Kap114p. The structural results obtained by implementing single-particle cryo-EM illustrate the determinants of the interaction between yTBP and the C-Terminal domain of Kap114p, revealing the critical residues involved in the binding. The interaction is further corroborated by other mutagenesis and Isothermal titration calorimetry (ITC) studies. Moreover biochemical analysis allowed to speculate that Kap114p dependent regulation pathways compete with other TBP sequestration pathways to confer salt tolerance to the yeast.

The results of this work have important implications for our understanding of the modulation of gene expression within nuclei and significantly contribute to improve our insight into the transcriptional regulatory network.

In general, the article is well written and I support the publication of this manuscript in *Nature Communication* after addressing the critical points listed below.

Minor points:

- For more clarity, I suggest the authors to indicate also the HEAT8 and HEAT13B loops in figure 1B, perhaps choosing different colors.

We thank the reviewer for this suggestion. We now indicate HEAT8B + loop and HEAT13B, and have colored them differently in Fig. 1. The color codes for HEAT8B + loop and HEAT13B have been changed accordingly throughout the revised manuscript and figures.

- In some figures (e.g., Fig. 2b) the text is overlapping with the data making it of difficult comprehension.

Our apologies for this issue. We have now corrected the figures containing the ITC results (Figs. 2, 4 and Extended Data Fig. 4).

- Please notice that some typos are present throughout the text. For example, in the ITC section page 25 line 22 you write “50mM of wild type or mutant Kap114p” and “500 mM of wild type or mutant yTBPC” but I think it should read 50 μ M and 500 μ M.

Our apologies for these typos. Now corrected.

- The concentrations used for ITC experiments are much higher than the Kd found by the authors for this interaction, therefore the error is also quite significant. It would be beneficial, if possible, to repeat the titration at lower concentrations (e.g., 8 μ M (cell) and 80 μ M(syringe)).

We thank the reviewer for his/her suggestion. We have now carried out a new set of ITC experiments to examine multiple WT and mutant Kap114p•yTBPC pairs at lower protein concentrations (Extended data Fig. 4b-g). However, in order to obtain an adequate signal-to-noise ratio in the data, the lowest concentrations of proteins we used were 25 (Kap114p; cell) and 250 (yTBPC; syringe) μ M, respectively. Nevertheless, under these conditions, the Kd values between Kap114p and yTBPC are still comparable to the values previously obtained at higher protein concentrations.

- At page 12 line 13 the authors claim that “simultaneous depletion of both kap114 and TAF1-TAND partially rescued the defective growth phenotypes caused by sole TAF1-TAND disruption, implying functional antagonism between these two TBP-interacting factors”. However, looking at Fig. 5a, it seems to me that both kap114 and TAF1-TAND depleted mutants display growth defects at high salt concentrations

and I would expect a much stronger effect for the mutant lacking both proteins, but this seems not to be the case. If possible, I would suggest the authors to provide further explanations within the text.

We thank the reviewer for his/her comments. Our previous study (Liao et al. 2020, PMID: 32484313) suggested that Kap114p serves as a suppressor to negatively regulate γ TBP-mediated transcription. Thus, knockout of *KAP114* in yeast resulted in a growth defect relative to wild-type strains under salt stress, as revealed by our spot assays (Fig. 5a-c). TAF1 is a TBP-associated factor. A handoff model has been proposed by which the TBP and TATA box DNA engagement facilitated by TAF proteins in the transcription factor IID is regulated. Generally, interactions between TAF1 and TBP are required for activated transcription in both yeast and mammalian cells (Martel et al. 2002, PMID: 11909971). TAF1 can facilitate activation of TBP-dependent transcription, and deletion of the TBP-interacting domain of TAF1 (Δ *taf1-tand*) in yeast also elicited a severe growth defect (Fig. 5a-c). As the reviewer points out, the *KAP114* and *TAF1-TAND* double deletion mutants did not show a much stronger growth defect. Instead, deletion of *KAP114* could “partially” rescue the growth defect caused by the *TAF1-TAND* single deletion mutant (Fig. 5a-c). Since Kap114p and TAF1 exert opposing transcriptional regulatory functions, we propose the existence of functional antagonism between these two TBP-interacting factors (Pages 12-13).

Moreover, under high salt conditions at 37 °C, the Δ *taf1-tand* growth defect masked the loss of *KAP114*, suggesting that *TAF1* is epistatic to *KAP114*, so *TAF1* may function downstream of *KAP114* under these conditions. Importantly, our results indicate that, under these conditions, Kap114p and TAF1 function in the same pathway that regulates TBP activity, even though Kap114p has other potential cargos (Page 17).

Please note that all spot assays were performed in triplicate (Figs. 5,6 and Extended Data Figs. 6,7). The spot assays shown in the main figures were further validated by growth curve measurements.

Major points:

• **The “materials and methods” section of the ITC data does not mention if reference experiments were acquired and subtracted from final data. However, this is a critical step for any ITC experiments and it is fundamental to be able to reproduce the results presented in the manuscript. Therefore, it is important to include reference experiments in the manuscript.**

We thank the reviewer for pointing this out. Reference ITC experiments (Kap114p (25 μ M) or γ TBP^C (250 μ M) versus buffer) are now included in our revised manuscript (Extended Data Fig. 4b,c). Essentially no heat change was observed in the reference experiments, so the final data shown in the manuscript was not reference-subtracted (please also see our Materials and Methods section, Page 30).

• **The ITC data presented are endothermic. This might be due to the presence of additional hydration. However, it seems that the complex is mainly characterized by the formation of hydrogen bonds and salt bridges, that usually are associated with exothermic processes. It will be useful to discuss a bit more in detail the ITC data on the base of the Cryo-EM structure and to include also the Δ H and Δ S values obtained using ITC in table1. In particular, the authors describe on page 9 line 4 that they did not observe any large conformational rearrangement upon complex formation, suggesting that there should be a favorable Δ S variation (positive Δ S) due to enhanced disorder in the environment. Including Δ S and Δ H data will be beneficial to further support and understand the structural data.**

As per the reviewer’s suggestion, we have now included the Δ H and Δ S values obtained using ITC in Extended Data Table 1. The positive Δ S indeed suggests an entropically-driven endothermic reaction for Kap114p and γ TBP^C assembly. Although multiple hydrogen bonds and salt bridges have been identified in the Kap114p \cdot γ TBP^C complex (protein-protein interactions), the endothermic reaction shown in the ITC data indicates the existence of protein-solvent interaction (e.g., water molecules), so energy is needed to break up the protein-solvent interaction (e.g., Kap114p and γ TBP^C to water) to facilitate Kap114p and γ TBP^C binding. These two opposing forces have been proposed as being a general feature of highly-charged complex assembly (Noskov & Lim 2001, PMID: 11463622). Thus, in this system, more energy is consumed to break “pre-existing” bonds (protein-solvent) than released to form bonds (protein-protein). Notably, no water molecules could be assigned in our cryo-EM structure presented in this study because it is at low resolution (~4 angstroms). This information has now been included in the revised manuscript (Page 19, lines 13-26).

Reviewer #2 (Remarks to the Author):

Liao et al. report a high resolution cryoEM structure of Kap114p bound to cargo TATA-box binding protein to reveal the structural basis of the high affinity binding of the cargo, as well as using yeast assays and structural guided mutants to study its transcriptional repressor function. Overall, the study is performed carefully, and the manuscript is written well. However, I do have a few comments that I believe will help improve the manuscript for publication.

Major concerns:

1. The Heat 19 loop is shown to be critical for binding and repressor function. Since its interaction is shown in the main figure, it would be good to show the density for this peptide on the map in supplementary so that readers can assess the confidence in the placement of the helix. It is hard to tell the density in extended data Figure 3A.

We thank the reviewer for this suggestion. We have now included a zoomed-in density map of the α -helix in the HEAT19 loop in our revised manuscript (Fig. 1b).

2. While the interaction interfaces are all investigated by mutagenesis, I feel that discussion is missing for the comparison of the interfaces and their contributions to affinity and function. Based on ITC mutagenesis, it seems like Heat8 interaction seems to be a lot more important than the HEAT 19 loop since the affinity dropped to micromolar range. The HEAT 19 loop deletion only drops the affinity to hundred of nanomolar. I wonder whether H19 loop deletion is still viable for import but the drop of affinity is sufficient to affect the transcriptional repression activity.

We thank the reviewer for these comments. We suggest that Kap114p without the HEAT19 loop can still deliver yTBP to the nucleus based on two pieces of evidence. First, as pointed out by the reviewer, deletion of the HEAT 19 loop only reduced the yTBP^C-binding affinity to a hundred nanomolar. However, this binding affinity is still greater than its counterparts that mediate nuclear import of TBP and show binding affinity to yTBP^C in the micromolar range (e.g., Kap95p and Kap121p) (Figs. 2, 4 and Extended Data Fig. 4 in this study and Liao et al. 2020, PMID: 32484313). Second, knockout of *KAP114* in yeast cells has been shown to cause partial mislocalization of yTBP to the cytoplasm (Morehouse et al. 1999, PMID: 10535958; Pemberton et al. 1999, PMID: 10385521). Our previous study (Liao et al. 2020, PMID: 32484313) also demonstrated that the cytoplasmic yTBP mislocalization due to *KAP114* knockout could be rescued by both the wild-type and HEAT19 loop deletion mutant proteins (Page 17).

Hence, the HEAT19 loop of Kap114p is not crucial for nuclear transport of yTBP. However, the HEAT19 loop deletion mutant could not rescue the growth defects of the *KAP114* knockout strain in high-salt conditions (Liao et al., 2020, PMID: 32484313). Therefore, together with our previous biochemical (e.g., EMSA) and transcriptomic (e.g., RNA-seq) analyses, in that study we proposed that Kap114p suppresses binding of yTBP to promoter DNA, thereby regulating gene transcription.

Here, based on our structural information, we show that the short α -helix in the Kap114p HEAT19 loop binds to the hydrophobic concave surface of yTBP^C in a similar fashion to a latch region of modifier of transcription 1 (Mot1). Notably, the latch deletion mutant of Mot1 still forms a complex with TBP, suggesting that it also minimally contributes to TBP binding (Wollmann et al. 2011, PMID: 21734658). Wollman et al. proposed that, instead of facilitating Mot1-TBP interaction, the latch region of Mot1 renders the TBP-DNA complex less stable, and thus enhances displacement of TBP from DNA. Likewise, our biochemical evidence shows that double point mutations of residues in the Kap114p•yTBP^C interacting surfaces (**HEAT8B + loop** (Kap114p(Y370A)+yTBP^C(N91A)) and **HEAT13B** (Kap114p(Q597A)+ yTBP^C(R141A)) greatly diminished binding between the Kap114p and yTBP^C components (Fig. 4e,i). In particular, the double mutant combining Y370A on Kap114p and N91A on yTBP^C resulted in a K_d value of ~1.5 μ M for the complex (Fig. 4e). However, the Kap114p deletion mutant lacking the α -helix of the **HEAT19 loop** (a.a. 932-941) had a binding affinity of ~150 nM to yTBP^C (Fig. 2b). These results suggest that HEAT8B + loop and HEAT13B mediate yTBP^C binding more significantly relative to the HEAT19 loop, and that the HEAT19 loop only partially contributes to the Kap114p and TBP interaction. However, our Chip-seq results have revealed that expression of wild-type *KAP114*, but not a HEAT19 loop deletion mutant, resulted in a substantial reduction of yTBP binding at the promoter DNA (Fig. 5e,f and Extended Data Fig. 6h,i), indicating suppression of yTBP-promoter interaction by Kap114p involves the HEAT19 loop of this latter. Taken together, our results support that the HEAT19 loop of Kap114p modulates TBP-mediated transcription, but that it is unlikely to be essential for the nuclear import of TBP. This information is now summarized in a paragraph in the Discussion section of our revised manuscript (Pages 17-19).

3. Since Kap114 is being compared to TAF1 and MOT1 as having similar interactions to yTBP, it would be informative to compare the binding affinities - either ITC measurement, or citing previous literature if available. This will also tie in with my last concern.

We thank the reviewer for this suggestion. Despite the K_d values being determined according to different approaches with buffer conditions that contained salt concentrations ranging from 100 to 500 mM, all of the results show that both Mot1p and TAF1 interact with TBP with a high affinity (~1 nM). A previous fluorescence anisotropy assay and EMSA revealed that Mot1p binds TBP with a high affinity and forms a relatively long-lived protein complex (K_d = ~1 nM, t_{1/2} = 12 minutes) (Gumbs et al., 2003, PMID: 12805227). Additionally, other fluorescence anisotropy studies have also shown that both TAF1-TAND (~100 amino acids) and full-length protein bind TBP with a high-affinity dissociation constant of ~1 nM (Bai et al, 1997, PMID: 9154807; Banik et al., 2001, PMID: 11677244). Therefore, not only do Kap114p, Mot1 and TAF1-TAND share similar structural features in terms of how they interact with TBP, but they also display high-affinity TBP binding in the sub-nanomolar range. Thus, Kap114p can be part of the TBP regulatory network, as it can compete with different TBP

interacting partners by binding to the same surface on TBP with a comparable binding affinity. This information has now been included in the Discussion section of our revised manuscript (Page 17, lines 14-16).

Minor concern:

1. Q597 can not be seen in Figure 4, but is a critical mutant that was used to validate modeling, perhaps another angle of the structure can be chosen to indicate the position of the residue.

We thank the reviewer for his/her suggestion. We have now included a different angle to illustrate the interactions of Kap114p HEAT13B with γ TBP^C and showing the Q597 residue (Extended Data Fig. 5g).

Reviewer #3 (Remarks to the Author):

The manuscript by Liao et al. reports the structure of the TATA-box binding protein (TBP) in interaction with the karyopherin Kap114. The authors also provide functional evidence that Kap114 might contribute to TBP promoter recruitment and regulatory roles in *Saccharomyces cerevisiae*. This study builds up on previous work by the same group that identified a role of Kap114 in the nuclear import of TBP. Here the authors used cryo-electron microscopy complemented with biochemical assays to thoroughly characterize Kap114-TBP interaction, revealing that Kap114 engages with TBP in similar manner as other important TBP regulators, including the TFIID subunit TAF1 and the Mot1 ATPase. Proliferation assays, genetic analyses and ChIP-seq analyses were performed to examine the effect of Kap114 on TBP function in vivo, but these experiments suffer from several flaws and are overinterpreted, making this work too preliminary for publication in its current form.

Specifically, I have the following concerns with the data presented in Figure 5 and Extended Figure 5 and how they are described in the text.

First there are major issues with the interpretation of the drop tests (Fig. 5a and Extended Fig. 5d).

- The conclusion that the Taf1-TAND antagonizes Kap114 for TBP function because the TAND deletion would rescue the loss of Kap114 is not supported by the data shown in Fig. 5a. The spot assays show, rather, that the Taf1-TAND and Kap114 regulate high salt tolerance independently because the double mutant displays intermediate growth as compared to each single mutant. In contrast, at 37°C, the Taf1-TAND growth defect is epistatic to the loss of Kap114, suggesting that the Taf1-TAND functions downstream of Kap114 in this condition.

We thank the reviewer for his/her comments. Our previous study (Liao et al. 2020, PMID: 32484313) suggested that Kap114p serves as a suppressor to negatively regulate γ TBP-mediated transcription. Thus, knockout of *KAP114* in yeast resulted in a growth defect relative to wild-type strains under salt stress, as revealed by our spot assays (Fig. 5a-c). TAF1 is a TBP-associated factor. A handoff model has been proposed by which the TBP and TATA box DNA engagement facilitated by TAF proteins in the transcription factor IID is regulated. Generally, interactions between TAF1 and TBP are required for activated transcription in both yeast and mammalian cells (Martel et al. 2002, PMID: 11909971). TAF1 can facilitate activation of TBP-dependent transcription, and deletion of the TBP-interacting domain of TAF1 (*Δtaf1-tand*) in yeast also elicited a severe growth defect (Fig. 5a-c). The *KAP114* and *TAF1-TAND* double deletion mutants did not show a much stronger growth defect. Instead, deletion of *KAP114* could “partially” rescue the growth defect caused by the *TAF1-TAND* single deletion mutant (Fig. 5a-c). Since Kap114p and TAF1 exert opposing transcriptional regulatory functions, we propose the existence of functional antagonism between these two TBP-interacting factors (Pages 12-13).

Moreover, under high salt conditions at 37 °C, the *Δtaf1-tand* growth defect masked the loss of *KAP114*, suggesting that *TAF1* is epistatic to *KAP114*, so *TAF1* may function downstream of *KAP114* under these conditions. Importantly, our results indicate that, under these conditions, Kap114p and TAF1 function in the same pathway that regulates TBP activity, even though Kap114p has other potential cargos (Page 17).

- Kap114 mutants that specifically abolishes its interaction with TBP are not tested, although the authors identified a specific region and the residues that are involved. Given the probable pleiotropy of kap114 mutant phenotypes, it is not possible to establish a direct link between the salt/heat stress sensitivity and the TBP binding role of Kap114 with a full kap114 deletion mutant.

We thank the reviewer for this comment. In our previous study (Liao et al. 2020, PMID: 32484313), we demonstrated that although deletion of the Kap114p HEAT19 loop (residues 899-956) does not perturb nuclear transport of γ TBP (Figure 3F), this mutant cannot rescue the yeast growth defects of *KAP114* knockout under high-salt conditions according to yeast spot assays (Figure 5B). These results support that the Kap114p- γ TBP interaction is correlated with salt sensitivity phenotypes, but it is not associated with the nuclear transport function.

Our ITC results also show that deletion of the HEAT19 loop (residues 932-941) reduces the binding constant between Kap114p and γ TBP^C to two orders of magnitude lower than WT Kap114p (~158 nM versus ~1 nM) (Fig. 2B and Extended Data table 1). However, single point mutations of Kap114p only elicited K_d values that are an order of magnitude greater than those of WT (e.g., Y370A (20 nM) and Q597A (7.4 nM)). Thus, single point

mutations in the large interface between Kap114p and γ TBP^C are unlikely to yield clear results by yeast spot assays.

However, we believe that the new data and control experiments suggested by the reviewers and that are now presented in the revised manuscript, together with our previously published data, are sufficient to support a link between the salt sensitivity and TBP binding role of Kap114p.

- Fig. 5a shows that the kap114 deletion mutant has a growth defect at 37C, thus under chronic heat stress, rather than osmotic stress, since the mutant strain grows as well as the wild-type control at 30C in 1.5M NaCl. The mutant might be sensitive to a combination of both stresses, but this is not tested explicitly. For this, the authors need to show spot assays performed in all four conditions (no stress, 1.5M NaCl at 30C, 37C alone, and 1.5M NaCl at 37C).

We thank the reviewer for this suggestion. In our revised manuscript, we now include spot assays showing that wild type and *KAP114* knockout strains display comparable growth rates under all four conditions recommended by the reviewer [No salt stress: Extended Data Fig. 6a (left and right panels; 30°C and 37°C); 1.5 M NaCl: Fig. 5 (left and right panels: 30°C and 37°C)]. We did examine *KAP114* knockout and wild-type strains under different stress conditions in our previous study (Liao et al. 2020, PMID: 32484313). The *KAP114* knockout and wild-type strains displayed comparable growth rates under many stress conditions (sorbital, acetic acid and MMS; Appendix Fig. S5 E in that study). However, the *KAP114* knockout strain only showed growth defects under high-salt conditions compared to the wild-type strain. These results suggest that the *KAP114* mutant is specifically sensitive to high salt, but not other stresses such as osmotic stress.

Notably, deletion of *KAP114* “partially” rescued the growth defects caused by the *TAF1-TAND* single deletion mutant only under high salt conditions, further corroborating genetic interaction between *KAP114* and *TAF1* under high salt conditions (Fig. 5a).

- A similar growth control is lacking for the Mot1 overexpression tests, which are spread between Fig. 5e and Extended Fig. 5d (consider to merge these panels into one figure for clarity). In addition, the effect of Mot1 on Kap114 must be tested against a strain that overexpresses Mot1 alone, as a control, to determine whether the growth defects induced by Mot1 overexpression depend on the loss of Kap114.

We thank the reviewer for these suggestions. We have now conducted new sets of spot assays including overexpression of *MOT1* alone as a control. The new data are shown in Fig. 6 of the revised manuscript, which represents a merger of Fig. 5e and Extended Data Fig. 5d, as recommended. Our results further corroborate that ectopic expression of *MOT1* in the *KAP114* knockout background elicits much stronger yeast growth defects (Fig. 6).

- Finally, it is imperative to confirm these spot assays, which are semi-quantitative, with an orthogonal approach to measure yeast proliferation, such as measuring cell concentration during the logarithmic phase of growth in each condition, with independent biological replicates. There are actually no details about how many independent replicates were performed for each spot assay.

We thank the reviewer for these comments. All spot assays were performed in triplicate (Figs. 5,6 and Extended Data Figs. 6,7) and this information is now mentioned in the respective figure legends. Furthermore, the spot assays shown in the main figures have been validated by growth curve measurements.

Second, there are important issues with the ChIP data. Specifically:

- The authors need to verify the levels of GFP-tagged TBP as well as of the WT and mutant Kap114 proteins using Western blotting, both in normal and high salt/heat stress conditions, to control for the putative differences in ChIP signal shown in Fig. 5c,d.

We thank the reviewer for this suggestion. We have performed immunoblotting assays using anti-GFP antibody to examine GFP-TBP levels under regular and stress conditions. Additionally, since *KAP114* is untagged and no antibody against Kap114p is available, we carried out qPCR to verify its expression levels. Essentially, our Western blot and qPCR results indicate that expression of both genes is comparable under the different conditions (Page 13, lines 14-25). These results have now been included in our revised manuscript (Extended Data Fig. 6d-g). Although we did detect slightly increased protein expression of GFP- γ TBP in the strain rescued by *KAP114*(Δ 899-956) (~1.2-fold relative to the strain rescued by *WT KAP114*) under high-salt conditions, our ChIP results for the *KAP114*(Δ 899-956)-rescued strain are comparable to that of the *KAP114* knockout strain rather than the WT-rescued strain (Fig. 5e,f and Extended Data Fig. 6h,i).

Notably, these results are consistent with our previously published data, even though the genetic background of the yeast strains is slightly different (Liao et al. 2020, PMID: 32484313; Appendix Figure S4). We detected comparable expression levels in terms of mRNA (qPCR) and proteins (Western blotting using FLAG tag) for wild-

type and HEAT19 loop-deleted Kap114 (driven by an endogenous promoter) in the *KAP114* knockout strain under regular and high-salt conditions.

- The authors need to provide evidence that GFP-tagged TBP is functional, particularly because its binding is much lower than expected at Pol II-dependent promoters (only 1480 genes identified).

We thank the reviewer for this comment and apologize for not providing a clearer explanation in the original manuscript. Though the yeast strain expressing GFP-tagged yTBP grew slower than its parental strain (BY4741, Extended Data Fig. 6b,c), the ChIP-seq experiments were carried out under the same conditions so the results should be comparable.

Moreover, we believe that GFP-tagged yTBP is functional as its chromatin binding ability is comparable to that of endogenous yTBP, as reported by Rhee and Pugh (PMID: 22258509). As shown in Figure 1 to the reviewer (see below), we observed significant GFP-yTBP occupancy at ~99% tRNA genes (272 out of 275 total tRNA genes in yeast) in the *KAP114* knockout strain without NaCl treatment. The three missing tRNA genes were absent from both input DNA and ChIP samples. Moreover, for Pol II-regulated mRNAs, we initially identified 1480 genes by setting a higher cutoff for the TBP ChIP-seq signals (promoters with a tag density of TBP ChIP-seq displaying a **3-fold change over input DNA**) in the *KAP114* knockout strain without NaCl treatment. For our new dataset, we analyzed the data for **2-fold enrichment** of TBP ChIP-seq signal (tag density > 100 rpkm) over input DNA in *Kap114* knockout cells (see Figure 1B to reviewer below), and identified 2,284 Pol II-regulated promoters. Notably, if we had applied 1.25-fold enrichment over input DNA for the TBP ChIP-seq signal, we would have identified 4395 Pol II-regulated promoters, which is comparable to the 4272 TBP-bound promoters of Pol II-regulated genes reported from a TBP ChIP-exo assay (tag density distributed from 2 to 1238, with a median of 33) by Rhee and Pugh (PMID: 22258509). Importantly, ~82% (3573 out of 4395) of our identified promoters overlap with those identified by TBP ChIP-exo. We now present two sets of results independently in the main figures (new dataset, with sequencing depth > 25 M reads/sample) and Extended Data figures (original dataset, with sequencing depth > ~7 M reads/sample), and have modified the manuscript text accordingly.

- The legend of Fig. 5c,d indicates that the data are represented as box-and-whisker plots, as seen on the graph. However, further down, the legend indicates that 'The results of three biological replicates are shown as mean \pm SD' - is this really what the whiskers are? There would tremendous variability then.

We apologize for this error, which has now been corrected.

- Using a two-tailed Student's t test to analyze the ChIP-seq data on Fig. 5c,d is not possible for several reasons, most importantly because it does not correct for multiple testing and, if I understood correctly that each dot represents one TBP peak, because these peaks do not correspond to independent replicates. Furthermore, the reporting summary indicates that ChIP-seq were done only once (1 replicate), precluding any differential analyses and interpretation of differences between strains and conditions.

We thank the reviewer for these comments and apologize for the oversights. We have now carried out a new transformation experiment and used a new colony to generate a new set of ChIP-seq data as a biological repeat. Then, we used a nonparametric Mann-Whitney U test to analyze the original and new datasets independently. Importantly, both datasets show consistent results, as determined by the new statistical analysis. We have included the new ChIP-seq results in main Fig. 5e,f and moved the original results to Extended Data Fig. 6h,i.

Minor concerns:

Page 2 Lane 4: Pol I transcribes rRNAs and Pol III transcribes tRNAs so they should be swapped in that sentence. Using 'respectively' requires to respect the order in each group of words.

Apologies for the confusion. This has now been corrected (Page 3, line 3).

Page 9: I do not really understand what Fig 2f is supposed to show. The data presented in Ext. 4 describe clearly that Taf1, Mot1, and Kap114 employ both similar and distinct structural features to interact with TBP. This part should be described in greater details in the text.

As recommended by the reviewer, we have modified Fig. 2f to show that TAF1-TAND, Mot1-latch, and Kap114 HEAT19 loop employ similar but distinct structural features to interact with TBP. Additionally, we have also included detailed information in the revised manuscript about how these motifs interact with the N-lobe of yTBP (Pages 9-10).

Page 9-10: The sentence that starts at the end of page 9 and ends on page 10 is confusing. It is difficult to understand which partner the authors are referring to, Kap114 or TBP, when describing the residues. Fig. 2i does not mention which protein sequences are shown, further complicating this sentence.

We apologize for the confusion. We have now clarified our text and Fig. 2i (the HEAT19 loop) accordingly.

Extended Fig. 5a lacks numbers for the y-axis and a schematic depiction of the locus shown, along with a scale bar for genomic distance.

Apologies for this oversight. Now corrected (Extended Data Fig. 7a).

REVIEWER COMMENTS

Reviewer #1 (Remarks to the Author):

All the critical points have been carefully addressed and clarified by the authors in the reviewed version of the manuscript, therefore I would support the publication of the present research work in Nature Communications.

Reviewer #2 (Remarks to the Author):

Liao et al. have revised the manuscript with updated controls and structural analysis, which I believe have improved the paper and will help readers understand the significance of its findings. I have one last point regarding the added discussion that I think needs to be corrected before the paper is accepted for publication.

In page 19, in the added discussion, it is said that no large conformational rearrangement of Kap114 is observed. However this is misleading - while the Kap114 solenoid did not have large conformational change, ordering of the h19loop helix when bound to TBP is a large change when considering all parts of the protein. The stabilization of the previously dynamic h19loop likely explains the entropically-driven TBP binding.

Reviewer #3 (Remarks to the Author):

I have now gone through the manuscript and appreciate the efforts made to address my concerns. The authors provided additional data and corrections that strengthen the manuscript, but not sufficiently to support publication. The functional aspect of the study remains too preliminary.

The new data are not strong enough to support a link between the salt sensitivity and TBP binding role of Kap114, beyond its role in nuclear import. The strongest evidence supporting this model has already been published by the authors in a previous paper (Liao et al. 2020, PMID: 32484313).

The data in this manuscript show that deletion of KAP114 causes a minor sensitivity to salt stress, if any. Fig. 5a shows a weak growth defect on plates, which severity seems to depend on the media used (see Figure 5A,B from Liao et al. 2020, PMID: 32484313). In contrast, Fig. 5b shows virtually identical growth rates and biomass production between WT and kap114 deletion mutant strains. From a strict genetic point of view, the partial rescue observed in the double kap114 taf1-TAND mutant can be interpreted in many distinct ways, including independent roles.

In severe stress conditions, combining high salt and heat shock, Kap114 clearly becomes important and functions upstream of Taf1 (Fig. 5), as I pointed out in my comment and acknowledged by the authors in their response. The same conclusion can be drawn for Mot1 (Fig. 6). Altogether, the strongest genetic evidence shown in this manuscript supports a model by which Kap114 imports TBP in the nucleus which is then loaded and unloaded from promoters by TFIID and Mot1, respectively, including in stress conditions.

The TBP ChIP-seq data still suffer from major flaws that make these data inconclusive because of a lack of statistical analyses, which is particularly critical when interpreting very small differences, and the use of a version of TBP that is only partially functional and which expression varies depending on Kap114.

- First, supplementary Fig. 6e and 6f contradict each other. The Western blot clearly shows less GFP-

TBP expression in both KAP114 and KAP114-899-956 rescue conditions, whereas the quantification shows no difference, maybe even a slight increase for the mutant. So it remains unclear whether Kap114 affects TBP levels, which is critical to interpret the ChIP-seq data.

- Second, ChIP is performed with a version of TBP that is evidently not fully functional, as shown in Extended Fig. 6b,c, regardless of its occupancy profile and the threshold used to quantify its binding genome-wide, so the results should be interpreted with extra caution. There are many examples of proteins that can ChIP efficiently without being able to carry out their functions.

- Third, the differences observed are small and not clearly described or quantified. I don't understand what 'substantial reduction' or 'slightly enhanced' TBP binding means, especially when the graphs shows at best a decrease of tag density from about 4.5- to 4-fold (a ~ 1.125 fold change) when comparing TBP binding at tRNAs in WT versus kap114 mutants in normal growth conditions.

- Fourth, the ChIP-seq experiments were now performed twice, which still precludes any statistical analysis of the reproducibility of the putative changes observed in the different strains and growth conditions. In fact, the Mann-Whitney U test used here is still not the right test. This non-parametric test is used to compare two groups. It is not appropriate to perform several Mann-Whitney tests comparing two groups at a time. For three groups or more, use the Kruskal-Wallis test followed by post-hoc comparisons. More importantly,, all tests assume that all the observations from both groups are independent of each other, which is not the case when using each mRNA or tRNA read as an observation.

Reviewer #1 (Remarks to the Author):

All the critical points have been carefully addressed and clarified by the authors in the reviewed version of the manuscript, therefore I would support the publication of the present research work in Nature Communications.

We thank the reviewer for his/her support to publish this work in *Nature Communications*.

Reviewer #2 (Remarks to the Author):

Liao et al. have revised the manuscript with updated controls and structural analysis, which I believe have improved the paper and will help readers understand the significance of its findings. I have one last point regarding the added discussion that I think needs to be corrected before the paper is accepted for publication.

In page 19, in the added discussion, it is said that no large conformational rearrangement of Kap114 is observed. However this is misleading - while the Kap114 solenoid did not have large conformational change, ordering of the h19loop helix when bound to TBP is a large change when considering all parts of the protein. The stabilization of the previously dynamic h19loop likely explains the entropically-driven TBP binding.

We thank the reviewer for making this point. Binding of γ TBP to Kap114p is an entropy-driven process because of the positive ΔS (Extended Data Table 1). Thus, energy is consumed in part to release ordered water molecules that bind to Kap114p to enable protein-protein bonds to form. In addition, a significant domain movement upon substrate binding has also been demonstrated to contribute to an entropy-driven process (Bezerra et al. 2011, PMID: 22493238). While no structural change was observed in the Kap114p solenoid, the HEAT19 loop between γ TBP-free and -bound Kap114p does undergo a large conformational change, additionally contributing to entropy-driven binding (Page 19-20).

Reviewer #3 (Remarks to the Author):

I have now gone through the manuscript and appreciate the efforts made to address my concerns. The authors provided additional data and corrections that strengthen the manuscript, but not sufficiently to support publication. The functional aspect of the study remains too preliminary.

The new data are not strong enough to support a link between the salt sensitivity and TBP binding role of Kap114, beyond its role in nuclear import. The strongest evidence supporting this model has already been published by the authors in a previous paper (Liao et al. 2020, PMID: 32484313).

We appreciate that the reviewer recognizes the importance of our previous publication. In the current manuscript, we provide structural detail on the Kap114p- γ TBP complex using cryo-EM, as well as show by ChIP-seq that Kap114p can modulate the interaction between γ TBP and promoter DNA. Thus, our manuscript greatly extends our previous publication and significantly strengthens the model proposed therein. Consequently, our proposed model is now supported by results/observations generated from different approaches and, importantly, is consistent in these two research articles. We do cite our previous publication in the current manuscript.

The data in this manuscript show that deletion of KAP114 causes a minor sensitivity to salt stress, if any. Fig. 5a shows a weak growth defect on plates, which severity seems to depend on the media used (see Figure 5A,B from Liao et al. 2020, PMID: 32484313). In contrast, Fig. 5b shows virtually identical growth rates and biomass production between WT and kap114 deletion mutant strains. From a strict genetic point of view, the partial rescue observed in the double kap114 taf1-TAND mutant can be interpreted in many distinct ways, including independent roles.

We thank the reviewer for these comments. The weak growth defect of the *KAP114* knockout strain in regular culture conditions is indeed associated with the media we used (Liao et al. 2020, PMID: 32484313, Figure 5A: YPD; Figure 5B: Dropout media; This study Figure 5a: Dropout media). However, the high-salt-induced defect we observed for *KAP114* knockout in yeast is independent of the media we used.

The spot assays shown in the main figures have been validated by growth curve measurements (Fig. 5a-c and 6a-e) as suggested by the reviewers in the original revision round. We now include *p* values to reinforce the significant differences in cell growth between WT and *KAP114* knockout strains under high-salt conditions at 30 °C and 37 °C (Figure 5b, c). These results clearly evidence the growth defects of the *KAP114* knockout strains compared to WT and are consistent with our spot assay results.

Indeed, the partial rescue observed for the double deletion mutants (*KAP114* and *TAF1-TAND*) could be interpreted in many distinct ways. However, based on the structural, biochemical, and transcriptomic analyses we

have presented in this study, we propose that biochemical and structural features of Kap114p likely enable it to function as part of the transcription regulatory network, in addition to its canonical function in nuclear import. This model is consistent with our previous publication (Liao et al. 2020, PMID: 32484313), which the reviewer acknowledges.

In severe stress conditions, combining high salt and heat shock, Kap114 clearly becomes important and functions upstream of TafI (Fig. 5), as I pointed out in my comment and acknowledged by the authors in their response. The same conclusion can be drawn for Mot1 (Fig. 6). Altogether, the strongest genetic evidence shown in this manuscript supports a model by which Kap114 imports TBP in the nucleus which is then loaded and unloaded from promoters by TFIID and Mot1, respectively, including in stress conditions.

We thank the reviewer for these comments. As the reviewer acknowledges, genetic results may be interpreted in many distinct ways. However, we do not propose a model based on results from a single approach. Instead, in this study, we have applied a multidisciplinary approach to examine the function of Kap114p. Surprisingly, our structural and biochemical evidence suggest that Kap114p is not likely to function solely as a nuclear transport receptor. In fact, structural features of Kap114p and its high binding affinity for yTBP may allow it to carry out functions beyond nuclear import. Moreover, together with our yeast genetics and TBP chromatin binding profiles, we propose that Kap114p acts as part of the transcription regulatory network by sequestering yTBP from promoter DNA binding. Please note that we do not exclude its function in the nuclear import pathway.

Knockout of *KAP114* in yeast cells has been shown previously to cause partial mislocalization of yTBP to the cytoplasm (Morehouse et al. 1999, PMID: 10535958; Pemberton et al. 1999, PMID: 10385521). Our previous study (Liao et al. 2020, PMID: 32484313) also demonstrated that this cytoplasmic yTBP mislocalization due to *KAP114* knockout could be rescued by both wild-type and HEAT19 loop deletion mutant proteins. Hence, the HEAT19 loop of Kap114p is not crucial for nuclear transport of yTBP. However, the HEAT19 loop deletion mutant could not rescue the growth defects of the *KAP114* knockout strain under high-salt conditions (Liao et al., 2020, PMID: 32484313). Based on our structural information, the HEAT19 loop binds to the hydrophobic concave surface of yTBP in a similar fashion to a latch region of Mot1 and TAF-TAND. Thus, multiple lines of evidence led us to propose our model that Kap114p acts in the transcription regulatory network, which we believe is reasonable and is now well-evidenced.

The TBP ChIP-seq data still suffer from major flaws that make these data inconclusive because of a lack of statistical analyses, which is particularly critical when interpreting very small differences, and the use of a version of TBP that is only partially functional and which expression varies depending on Kap114.

- First, supplementary Fig. 6e and 6f contradict each other. The Western blot clearly shows less GFP-TBP expression in both *KAP114* and *KAP114-899-956* rescue conditions, whereas the quantification shows no difference, maybe even a slight increase for the mutant. So it remains unclear whether Kap114 affects TBP levels, which is critical to interpret the ChIP-seq data.

We thank the reviewer for these comments. We have now repeated in triplicate the Western blot analysis using anti-GFP antibodies to carefully examine levels of GFP-yTBP in the three strains under regular and high-salt conditions. Quantification revealed no statistical differences among the new and previous datasets (six repeats) (Extended data Fig. 6f-h). This outcome is further validated by our new RT-qPCR experiments, in which we examined yTBP expression at the transcriptional level (Extended data Fig. 6i).

- Second, ChIP is performed with a version of TBP that is evidently not fully functional, as shown in Extended Fig. 6b,c, regardless of its occupancy profile and the threshold used to quantify its binding genome-wide, so the results should be interpreted with extra caution. There are many examples of proteins that can ChIP efficiently without being able to carry out their functions.

We thank the reviewer for these comments. To further compare the functionality of GFP-tagged and endogenous TBP, we have carried out new RT-qPCR experiments to examine the gene expression mediated by RNA Pol I (*RDN58*), RNA Pol III (*SNR6*), and RNA Pol II (TFIID-dependent *RPS5*, *RPS8A*, *RPS3*; SAGA-dependent *PYK1* and *PGK1*) in the strain expressing GFP-tagged yTBP and its parental strain (BY4741). Notably, we detected no difference in expression for the respective genes under regular and high-salt conditions (Extended data Fig. 6d,e). Given that TBP is involved in the transcriptional activity of RNA Pol I, II, and III, these results strongly support that the GFP tag on yTBP does not affect its function in transcription. Hence, we are confident that GFP-tagged yTBP not only can be "ChIPed" efficiently, as the reviewer acknowledges, but can also carry out its transcriptional function as efficiently as the wild-type protein. We have now revised the manuscript accordingly to reinforce this point (Page 13).

For our RT-qPCR analysis (new experiments), we selected 18 additional genes and examined their expression in *KAP114* knockout yeast strains expressing yTBP with or without the GFP tag (Extended data Fig. 8a-f). The highly correlated gene expression patterns for these 18 genes across all *KAP114*-knockout strains also supports that the GFP tag does not affect TBP-mediated transcription. Notably, expression of *ACTIN*, the internal control gene, may also be modulated by yTBP. Thus, we conducted a pairwise comparison between strains expressing

yTBP with and without the GFP tag within **each** of the *KAP114* knockout strains, which had been rescued by vector (control), WT or mutant *KAP114* (Extended data Fig. 8a-f).

Notably, the GFP-tagged yTBP strain was created by the O'Shea and Weissman groups (Huh et al., 2003, PMID: 14562095). As reported in that respective paper, GFP-tagged yTBP displayed a nuclear localization in yeast. Moreover, *KAP114* knockout promotes partial mislocation of GFP-yTBP to the nucleus (Liao et al., 2020, PMID: 32484313), consistent with findings for HA- and protein A-tagged yTBP (Morehouse et al., PMID: 10535958; Pemberton et al., PMID: 10385521). Thus, the GFP tag does not interfere with yTBP binding to its binding partners to facilitate nuclear transport. Importantly, in the current study, we demonstrate that yTBP uses the same binding mode to interact with Kap114p and other TBP-associated proteins (e.g., TAF1-TAND and Mot1). Hence, GFP-tagged TBP should be able to interact with transcription regulatory factors, and it behaves the same way as the wild-type protein in nuclear import and transcriptional regulation.

The GFP-tagged yTBP strain was created by oligonucleotide-directed homologous recombination to insert the tag in the chromosomal location of the open reading frame (Huh et al., 2003, PMID: 14562095). While the tag does not disrupt the gene's upstream regulatory or promoter sequences, it cannot be excluded that integration of the cassette (tag and selection markers) alters cell proteomics or metabolomics (Coumans et al., 2014, PMID: 23899627; Bordet et al., 2022, PMID: 35456831). Thus, that the strain expressing GFP-tagged yTBP grew more slowly than its parental strain (BY4741, Extended Data Fig. 6b,c) could be due to indirect effects on pathways other than nuclear transport and transcriptional regulation.

Importantly, currently no good antibody for “yeast” TBP (e.g., ChIP grade) is commercially available. Thus, most functional studies on TBP in yeast have been carried out using antibodies against tags (e.g. GFP and HA). Nevertheless, the ChIP-seq experiments we carried out in the current study were conducted under the same conditions (e.g. culture media, genomic background, and nearly identical levels of GFP-yTBP or Kap114p), so the results should be comparable.

- Third, the differences observed are small and not clearly described or quantified. I don't understand what 'substantial reduction' or 'slightly enhanced' TBP binding means, especially when the graphs shows at best a decrease of tag density from about 4.5- to 4-fold (a ~1.125 fold change) when comparing TBP binding at tRNAs in WT versus kap114 mutants in normal growth conditions.

We apologize for the confusion. As per the reviewer's suggestions, we have now performed a Kruskal-Wallis test with Dunn's post-hoc test on the two ChIP-seq datasets (Fig. 5e,f and Extended data Fig. 7a,b). The *p* values for the pairwise comparisons are annotated by asterisks in the figures and we list details in Tables 1 and 2 for the reviewer (see below). Even though the graphs show a small difference between the pairs (most likely due to a large number of genes being included: 2,284 mRNA/272 tRNA in dataset 2 and 1,480 mRNA/272 tRNA in dataset 1), the *p* values clearly indicate their statistical differences. We have now replaced the terms “substantial reduction” and “slightly enhanced” with “statistically significant reduction” and “statistically significant enhancement”, respectively, in the revised manuscript (Highlighted in yellow, pages 14-15)

- Fourth, the ChIP-seq experiments were now performed twice, which still precludes any statistical analysis of the reproducibility of the putative changes observed in the different strains and growth conditions. In fact, the Mann-Whitney U test used here is still not the right test. This non-parametric test is used to compare two groups. It is not appropriate to perform several Mann-Whitney tests comparing two groups at a time. For three groups or more, use the Kruskal-Wallis test followed by post-hoc comparisons. More importantly,, all tests assume that all the observations from both groups are independent of each other, which is not the case when using each mRNA or tRNA read as an observation.

We thank the reviewer for these comments. We have now carried out a Kruskal-Wallis test with Dunn's post-hoc test for the two ChIP-seq datasets (Fig. 5e,f and Extended data Fig. 7a,b). The *p* value from the Kruskal-Wallis test for each group is shown in the figures and supports statistical significance. Significant pairwise comparisons are annotated by asterisks in the figures. Additionally, the *p* values for each pair after Dunn's post-hoc test are detailed in Tables 1 and 2 below for the reviewer. Notably, the results of the Kruskal-Wallis and Mann-Whitney U tests are identical, supporting the robustness of our data, regardless of the statistical method used to analyze it.

Furthermore, we also now present scatterplots to illustrate the correlation in fold-change for pairwise samples from ChIP-seq repeats 1 and 2 under regular and high-salt conditions (Extended data Fig. 7c-n). Pearson correlation coefficients for each pair indicate that the ChIP-seq results display high (mRNA) or moderate (tRNA) reproducibility. The tRNA results present moderate reproducibility even though Kap114p modulates binding of yTBP to 272 out of 275 SDG-annotated tRNA genes in both datasets. Moreover, the overall profiles of the box-whisker plots of these two datasets are comparable. Thus, the observed patterns are most likely due to the fold-change differences in these 272 tRNA genes between the two datasets.

Table 1 to reviewer |Dunn's test of ChIP-Seq data

ChIP-Seq repeat 1		p value	Shown in Supplemental Fig. 7a
mRNA			
Comparison			
Dropout media			
$\Delta kap114$	vs $\Delta kap114 + KAP114$ WT	1.14E-171	****
$\Delta kap114$	vs $\Delta kap114 + KAP114 (\Delta 899-956)$	6.09E-59	
$\Delta kap114 + KAP114$ WT	vs $\Delta kap114 + KAP114 (\Delta 899-956)$	7.74E-32	****
Dropout media + 1.5 M NaCl			
$\Delta kap114$	vs $\Delta kap114 + KAP114$ WT	0.92	n.s.
$\Delta kap114$	vs $\Delta kap114 + KAP114 (\Delta 899-956)$	8.14E-11	
$\Delta kap114 + KAP114$ WT	vs $\Delta kap114 + KAP114 (\Delta 899-956)$	4.16E-11	****
Between regular and high salt conditions			
$\Delta kap114$	vs $\Delta kap114$ (1.5 NaCl)	9.04E-171	
$\Delta kap114$	vs $\Delta kap114 + KAP114$ WT (1.5 NaCl)	1.47E-169	
$\Delta kap114$	vs $\Delta kap114 + KAP114 (\Delta 899-956)$ (1.5 NaCl)	1.21E-258	
$\Delta kap114 + KAP114$ WT	vs $\Delta kap114$ (1.5 NaCl)	0.94	
$\Delta kap114 + KAP114$ WT	vs $\Delta kap114 + KAP114$ WT (1.5 NaCl)	0.87	n.s.
$\Delta kap114 + KAP114$ WT	vs $\Delta kap114 + KAP114 (\Delta 899-956)$ (1.5 NaCl)	1.33E-10	
$\Delta kap114 + KAP114 (\Delta 899-956)$	vs $\Delta kap114$ (1.5 NaCl)	3.49E-31	
$\Delta kap114 + KAP114 (\Delta 899-956)$	vs $\Delta kap114 + KAP114$ WT (1.5 NaCl)	6.00E-31	
$\Delta kap114 + KAP114 (\Delta 899-956)$	vs $\Delta kap114 + KAP114 (\Delta 899-956)$ (1.5 NaCl)	9.60E-74	

ChIP-Seq repeat 1		p value	Shown in Supplemental Fig. 7b
tRNA			
Comparison			
Dropout media			
$\Delta kap114$	vs $\Delta kap114 + KAP114$ WT	3.79E-21	****
$\Delta kap114$	vs $\Delta kap114 + KAP114 (\Delta 899-956)$	0.004	
$\Delta kap114 + KAP114$ WT	vs $\Delta kap114 + KAP114 (\Delta 899-956)$	8.70E-35	****
Dropout media + 1.5 M NaCl			
$\Delta kap114$	vs $\Delta kap114 + KAP114$ WT	1.29E-08	****
$\Delta kap114$	vs $\Delta kap114 + KAP114 (\Delta 899-956)$	1.30E-06	
$\Delta kap114 + KAP114$ WT	vs $\Delta kap114 + KAP114 (\Delta 899-956)$	6.52E-26	****
Between regular and high salt conditions			
$\Delta kap114$	vs $\Delta kap114$ (1.5 NaCl)	7.29E-103	
$\Delta kap114$	vs $\Delta kap114 + KAP114$ WT (1.5 NaCl)	1.45E-56	
$\Delta kap114$	vs $\Delta kap114 + KAP114 (\Delta 899-956)$ (1.5 NaCl)	2.67E-153	
$\Delta kap114 + KAP114$ WT	vs $\Delta kap114$ (1.5 NaCl)	1.10E-33	
$\Delta kap114 + KAP114$ WT	vs $\Delta kap114 + KAP114$ WT (1.5 NaCl)	1.46E-10	****
$\Delta kap114 + KAP114$ WT	vs $\Delta kap114 + KAP114 (\Delta 899-956)$ (1.5 NaCl)	2.43E-64	
$\Delta kap114 + KAP114 (\Delta 899-956)$	vs $\Delta kap114$ (1.5 NaCl)	1.71E-131	
$\Delta kap114 + KAP114 (\Delta 899-956)$	vs $\Delta kap114 + KAP114$ WT (1.5 NaCl)	3.87E-78	
$\Delta kap114 + KAP114 (\Delta 899-956)$	vs $\Delta kap114 + KAP114 (\Delta 899-956)$ (1.5 NaCl)	6.08E-188	

Table 2 to reviewer|Dunn's test of ChIP-Seq data

ChIP-Seq repeat 2

mRNA	Comparison	p value	Shown in Fig. 5e
Dropout media			
$\Delta kap114$	vs $\Delta kap114 + KAP114 WT$	1.32E-11	****
$\Delta kap114$	vs $\Delta kap114 + KAP114 (\Delta 899-956)$	0.002	
$\Delta kap114 + KAP114 WT$	vs $\Delta kap114 + KAP114 (\Delta 899-956)$	5.38E-2	****
Dropout media + 1.5 M NaCl			
$\Delta kap114$	vs $\Delta kap114 + KAP114 WT$	1.29E-26	****
$\Delta kap114$	vs $\Delta kap114 + KAP114 (\Delta 899-956)$	2.87E-10	
$\Delta kap114 + KAP114 WT$	vs $\Delta kap114 + KAP114 (\Delta 899-956)$	1.08E-64	****
Between regular and high salt conditions			
$\Delta kap114$	vs $\Delta kap114 (1.5 NaCl)$	5.01E-201	
$\Delta kap114$	vs $\Delta kap114 + KAP114 WT (1.5 NaCl)$	2.63E-85	
$\Delta kap114$	vs $\Delta kap114 + KAP114 (\Delta 899-956) (1.5 NaCl)$	1.38E-292	
$\Delta kap114 + KAP114 WT$	vs $\Delta kap114 (1.5 NaCl)$	5.75E-122	
$\Delta kap114 + KAP114 WT$	vs $\Delta kap114 + KAP114 WT (1.5 NaCl)$	1.50E-37	****
$\Delta kap114 + KAP114 WT$	vs $\Delta kap114 + KAP114 (\Delta 899-956) (1.5 NaCl)$	5.11E-195	
$\Delta kap114 + KAP114 (\Delta 899-956)$	vs $\Delta kap114 (1.5 NaCl)$	5.27E-244	
$\Delta kap114 + KAP114 (\Delta 899-956)$	vs $\Delta kap114 + KAP114 WT (1.5 NaCl)$	6.84E-114	
$\Delta kap114 + KAP114 (\Delta 899-956)$	vs $\Delta kap114 + KAP114 (\Delta 899-956) (1.5 NaCl)$	$p \rightarrow 0$	

ChIP-Seq repeat 2

tRNA	Comparison	p value	Shown in Fig. 5f
Dropout media			
$\Delta kap114$	vs $\Delta kap114 + KAP114 WT$	2.26E-11	****
$\Delta kap114$	vs $\Delta kap114 + KAP114 (\Delta 899-956)$	4E-05	
$\Delta kap114 + KAP114 WT$	vs $\Delta kap114 + KAP114 (\Delta 899-956)$	3.21E-02	****
Dropout media + 1.5 M NaCl			
$\Delta kap114$	vs $\Delta kap114 + KAP114 WT$	1.02E-10	****
$\Delta kap114$	vs $\Delta kap114 + KAP114 (\Delta 899-956)$	6.28E-05	
$\Delta kap114 + KAP114 WT$	vs $\Delta kap114 + KAP114 (\Delta 899-956)$	1.25E-25	****
Between regular and high salt conditions			
$\Delta kap114$	vs $\Delta kap114 (1.5 NaCl)$	7.00E-105	
$\Delta kap114$	vs $\Delta kap114 + KAP114 WT (1.5 NaCl)$	9.48E-53	
$\Delta kap114$	vs $\Delta kap114 + KAP114 (\Delta 899-956) (1.5 NaCl)$	3.12E-146	
$\Delta kap114 + KAP114 WT$	vs $\Delta kap114 (1.5 NaCl)$	2.90E-51	
$\Delta kap114 + KAP114 WT$	vs $\Delta kap114 + KAP114 WT (1.5 NaCl)$	8.11E-18	****
$\Delta kap114 + KAP114 WT$	vs $\Delta kap114 + KAP114 (\Delta 899-956) (1.5 NaCl)$	5.09E-81	
$\Delta kap114 + KAP114 (\Delta 899-956)$	vs $\Delta kap114 (1.5 NaCl)$	1.53E-147	
$\Delta kap114 + KAP114 (\Delta 899-956)$	vs $\Delta kap114 + KAP114 WT (1.5 NaCl)$	7.06E-84	
$\Delta kap114 + KAP114 (\Delta 899-956)$	vs $\Delta kap114 + KAP114 (\Delta 899-956) (1.5 NaCl)$	4.85E-196	

Reviewer #3

Specific answers to the authors' responses:

However, the high-salt-induced defect we observed for KAP114 knockout in yeast is independent of the media we used:

I do not understand this statement. The legend to Fig. 5A,B of Liao et al. 2020, PMID: 32484313, states that the strains were diluted serially and spotted onto YPD (5A) or minimal (5B) media in the presence of 1.5 M NaCl. The *kap114* deletion mutant does not show any obvious defect on Fig. 5A (YPD) whereas it does on Fig. 5B (minimal media).

We thank the reviewer's comments. In Fig. 5A, B of Liao et al. 2020, PMID: 32484313, we did mention that wild-type and KAP114 knockout strains were diluted serially and spotted onto YPD (Fig. 5A) and minimal media (Fig. 5B) plates in the presence of 1.5 M NaCl at 30 and 37 °C. However, we did not have statements that "the KAP114 deletion mutant does not show any obvious defect on YPD, whereas it does on minimal media". Instead, in the main text, we described clearly that the KAP114 knockout strain grew much more slowly in high-salt conditions compared to the wild-type strain in the YPD and minimal media, especially at 37 °C (Fig. 5A and B). The KAP114 deletion mutant showed a severe growth defect on minimal media. However, the cell growth revealed by the yeast spot assays showed the same trend in the YPD and minimal media.

We now include p values to reinforce the significant differences in cell growth between WT and KAP114 knockout strains under high-salt conditions at 30C and 37C (Figure 5b, c):

These P values suggest a difference at late time points at 30C, at day 5, three days after reaching stationary phase. This would suggest a minor defect in maintaining viability in starved conditions, not a growth defect. Rather, the WT and mutant show virtually identical exponential growth and total biomass production at 30C. This has to be acknowledged and described accurately.

We thank the reviewer for his/her comments. The WT and *KAP114* knockout mutants showed minor differences in the growth rate during the exponential stage and the final total biomass at 30°C. However, The differences were much more evident at 37°C under high salt conditions. As the reviewer suggested, we cannot exclude the possibility that *KAP114* knockout also can lead to growth defects in the cell viability in addition to growth rate. We have included this statement in the revised manuscript (Pages 12-13).

Otherwise, my concerns about all the CHIP data has been addressed appropriately and I thank the authors for their efforts to clarify this points and provide the right controls to support their conclusions.

We thank the reviewer for the constructive input to make this manuscript better.